# Variational Variance:
# Simple and Reliable Noise Variance Parameterization

## Abstract

Models employing heteroscedastic Gaussian likelihoods parameterized by amortized mean and variance networks are both probabilistically interpretable and highly flexible, but unfortunately can be brittle to optimize. Maximizing log likelihood encourages local Dirac densities for sufficiently flexible mean and variance networks, especially for data points lacking nearby neighbors. Gradients near these unbounded optima explode, prohibiting convergence of the mean and thus requiring high noise variance to explain the dependent variable. We propose posterior predictive checks to identify such failures, which we observe can surreptitiously occur alongside high model likelihoods. We find existing approaches that bolster optimization of mean and variance networks to improve likelihoods still exhibit poor predictive mean and variance calibration. Our simpler solution, to treat heteroscedastic variance variationally in an Empirical Bayes regime, regularizes variance away from zero and stabilizes optimization, allowing us to preserve or outperform existing likelihoods while improving predictive mean and variance calibrations and thereby sample quality. We empirically demonstrate these findings on a variety of regression and variational autoencoding tasks.

## 1 Introduction

While reliable uncertainty estimation has long been important to machine learning in the forms of *active learning* (Cohn et al., 1996) and *reinforcement learning* (Ghavamzadeh et al., 2016), improving predictive uncertainty estimates often is overlooked in favor of achieving state-of-the-art predictive mean and mode estimatesValdenegro-Toro (2021). Bayesian uncertainty is comprised of *epistemic* (model) and *aleatoric* (data) uncertainties (Kendall & Gal, 2017), both of which a model's predictive distribution ideally captures. Sicking et al. (2021) broadly categorize predictive uncertainty estimation into Bayesian approximations (e.g. Monte Carlo dropout (Gal & Ghahramani, 2016)), ensemble approaches (Lakshminarayanan et al., 2016), and parametric models that output heteroscedastic variance or covariance estimates (Nix & Weigend, 1994). This article focuses on the latter: modeling conditional $Y|X = x$ as a heteroscedastic Gaussian $\mathcal{N}(Y|\mu(x), \sigma^2(x))$ parameterized by mean network $\mu(\cdot)$ and variance network $\sigma^2(\cdot)$. Optimizing $\mu(\cdot)$ and $\sigma^2(\cdot)$ simultaneously using maximum likelihood estimation (MLE), however, can be unstable (Takahashi et al., 2018). For sufficiently flexible networks, maximizing the log likelihood simultaneously encourages $\mu(x_i) \to y_i$ and $\sigma^2(x_i) \to 0$. The fact that $\sigma^{-2}(x_i)$ appears as a multiplicative factor in the mean's gradient underlies why jointly optimizing $\mu(\cdot)$ and $\sigma^2(\cdot)$ can be brittle: minuscule errors by $\mu(x_i)$ produce inappropriately large parameter updates. Takahashi et al. (2018) refer to this undesirable tendency as the "zero variance" problem. In essence, the variance network increases the learning rate of the mean network as it improves, directly opposing stochastic gradient descent convergence criteria (Robbins & Monro, 1951).

Takahashi et al. (2018) resolve optimization instability in variational autoencoders (VAEs) (Kingma & Welling, 2013; Rezende et al., 2014) by reparameterizing the observed data distribution from a Normal likelihood to a Student's $t$. Experimentally, they demonstrate this proposal smooths out the log likelihood objective's learning curves and results in higher log likelihoods. Their experiments, however, do not assess the quality of their models' predictive mean and variance. Our experiments, where we also extend their proposal to regression tasks, show that despite achieving high log likelihoods, the Student's $t$ parameterization does not

reliably produce well calibrated mean and variance estimates. Because local convergence failures of the mean can be explained away with noise variance, models can attain high log likelihoods even while making poorly calibrated predictive mean and variance estimates for underrepresented groups (e.g. algorithmic fairness and racial bias). Detlefsen et al. (2019) recognize that if $x_i$ has nearby neighbor $x_j$ and the mean network is not sufficiently flexible to become arbitrarily close to both $y_i$ and $y_j$ simultaneously, then the "zero variance" problem (Takahashi et al., 2018) is avoided since placing Dirac densities on both $Y = y_i | X = x_i$ and $Y = y_j | X = x_j$ is no longer an option. Thus, data with meaningful neighbors can eliminate local conditions for instability and allow the model to produce sensible local mean and variance estimates that capture underlying aleatoric uncertainty. To this end, they propose a "locality sampler" that ensures $x_i$'s $k$ nearest neighbors accompany $x_i$ in a batch. This proposal is in addition to three others (see appendix C.1 for a discussion), one of which is also reparameterizing the Normal likelihood to a Student's $t$. Applied to regression and VAEs, their proposals seem to improve log likelihood and mean calibration. They examine variance calibration only when they have access to the true variance values.

To a large extent, Detlefsen et al. (2019) rely on nearby neighbors to solve the "zero variance" problem. Their proposal, in addition to their Student's $t$ reparameterization, offered the most benefit in their ablation studies. Unfortunately, there is no guarantee that all datapoints in a dataset will have meaningful neighbors that can help avoid local "zero variance" problems. Lacking stabilizing neighbors is severely exacerbated in higher dimensions. Increasing dimensionality even moderately (10-15) can make Euclidean distances between a point and its nearest and furthest neighbors indistinguishable (Beyer et al., 1999). Indeed, we find that the proposals of Detlefsen et al. (2019) breakdown for high-dimensional VAEs.

**Contributions.** Herein we propose a previously unrecognized, attractively simple, and probabilistically principled solution to improve predictive mean and variance calibration of heteroscedastic Gaussian densities without requiring data to have nearby neighbors. We treat heteroscedastic noise variance variationally and allow appropriately selected priors to counteract variance's tendency towards zero when mean errors are small. We eliminate the need for tuning priors by adopting an Empirical Bayes perspective, which allows prior parameters to be optimized.

Additionally, we employ posterior predictive checks (PPCs) (Gelman et al., 2013) to critique predictive mean and variance calibrations. PPCs posit a well-fit, appropriate model should, with high probability, produce new data that looks similar to the observed data. A common PPC is to evaluate the log predictive likelihood on held-out test data, but this unfortunately does not test mean and variance calibrations. Alternatively, sampling values from the predictive distribution and looking for systematic discrepancies with the original data will test both mean and variance calibrations. We use multiple PPCs to critique *both* a model's log likelihood *and* its predictive mean and variance. Notably, our usage of PPCs does *not* require knowing the true underlying variance.

To summarize, our contributions are:

- **(Primary)** To the best of our knowledge, we are the first to propose a variational treatment of noise variance as a probabilistically principled method to improve both predictive mean and variance calibration for heteroscedastic Gaussian models that use neural network parameter maps.

- **(Secondary)** We explore which priors on the variance provide the best performance, including several novel priors we introduce and optimize via Empirical Bayes.

- **(Secondary)** We advocate for and demonstrate the benefit of using PPCs to test if a heteroscedastic model produces well-calibrated predictive mean and variance estimates.

Section 2 formalizes our variational treatment of noise variance. Sections 3 and 4 respectively apply our proposals to regression and VAEs.

## 2 Variational Empirical Bayes for Noise Variance

We propose reparameterizing local heteroscedastic Gaussian likelihoods from $\mathcal{N}(Y_i | \mu(x_i), \sigma^2(x_i))$ to $\mathcal{N}(Y_i | \mu(x_i), \lambda_i)$ using local latent precision $\lambda_i$ rather than variance for computational convenience. We

then place a prior over local precisions and perform variational inference (VI) (Blei et al., 2017). In our setting, VI posits local variational family $q(\lambda_i|\alpha(x_i), \beta(x_i)) \triangleq \Gamma(\lambda_i; \alpha(x_i), \beta(x_i))$ to approximate the true posterior $p(\lambda|\mathcal{D}) \approx \prod_i q(\lambda_i|\alpha(x_i), \beta(x_i))$, where $\mathcal{D} = \{(x_i, y_i)\}_{i=1}^{N}$. Using neural networks $\alpha(\cdot)$ and $\beta(\cdot)$ to parameterize the variational Gamma distribution is an example of *amortized variational inference* (Kingma & Welling, 2013). Our amortized VI setup minimizes the variational posterior's Kullback–Leibler (KL) divergence from the true posterior by maximizing the evidence lower bound (ELBO or $\mathcal{L}$ for short),

$$\sum_{i=1}^{N} \mathbb{E}_q \left[ \log \mathcal{N}(Y_i = y_i|\mu(x_i), \lambda_i) \right] - D_{KL}\big(q(\lambda_i|\alpha(x_i), \beta(x_i)) \mid\mid p(\lambda_i)\big). \tag{1}$$

The expected log likelihood evaluates analytically (appendix eq. (5)). The KL divergence in the ELBO provides a probabilistically principled way to regularize variance away from pathological zeros. A prior that has negligible mass for near-zero variances but sufficient entropy for reasonable non-zero variances will both regulate variance away from zero for data without neighbors while also allowing the true noise variance to be learned for data with neighbors. Because precision is local, we maintain a 1:1 ratio of local log likelihoods to KL divergences. Thus, the regularization effect away from zero variance for data without neighbors is unaffected by dataset size. Equation (1) treats $\mu(\cdot)$'s optimization ambiguously since there are several valid interpretations. The simplest is that we are performing hybrid inference with MLE for the mean and Variational Bayes for the precision. A fully Variational Bayesian interpretation of eq. (1) is that we place a uniform prior over the reals on $\mu_i$ and define $q(\mu_i) \triangleq \delta(\mu(x_i))$. The mean's resulting KL divergence subtracts a constant, negative infinity from the ELBO, which can be ignored since it has zero gradient w.r.t. to the mean network's parameters. From an Empirical Bayes perspective, eq. (1) arises when we define $p(\mu_i) \triangleq \delta(\mu(x_i))$ and $q(\mu_i) \triangleq \delta(\mu(x_i))$. Here, the mean's KL term vanishes since the prior and variational distributions are identical, and we simultaneously optimize their shared parameters.

After optimization, we turn our attention to making predictions on some newly observed covariate $x_* \notin \mathcal{D}$, for which we define the *variational posterior predictive*

$$p(Y_*|X_* = x_*, \mathcal{D}) = \int \mathcal{N}(Y_*|\mu(x_*), \lambda)q(\lambda|\alpha(x_*), \beta(x_*))d\lambda = \text{T}(Y_*|\mu(x_*), \nu(x_*), \lambda(x_*)), \tag{2}$$

the expectation of the reparameterized likelihood w.r.t. our variational posterior. Because we define $q(\lambda_i|\alpha(x_i), \beta(x_i)) \triangleq \Gamma(\lambda_i; \alpha(x_i), \beta(x_i))$, integration analytically admits a Student's $t$ with degrees-of-freedom $\nu(x_*) \equiv 2\alpha(x_*)$ and precision $\lambda(x_*) \equiv \frac{\alpha(x_*)}{\beta(x_*)}$ (Bishop (2006) derives this result in section 2.3.7). If the variational posterior reasonably approximates the true posterior, integration over $\lambda$ accounts for epistemic uncertainty. Conditioning on $\mathcal{D}$ is absent in the integrand; it is implicit since the network parameters for $\mu(\cdot)$, $\alpha(\cdot)$, and $\beta(\cdot)$ were fit using $\mathcal{D}$. We therefore abbreviate the *variational posterior predictive* as $p(Y_*|X_*)$.

We consider the homoscedastic $p(\lambda_i)$ and heteroscedastic $p(\lambda_i|x_i)$ priors in table 1, but use $p(\lambda_i)$ to generally refer to both throughout this article (e.g. eq. (1)). Because of our chosen variational family, $q(\lambda|\alpha(x), \beta(x))$, our variational posterior predictive is always heteroscedastic. Thus, we really only care about which prior(s) offer optimal PPC performance. That said, having heteroscedasticity exist both in the generative process and in inference may be philosophically preferable.

We test the standard conjugate **Gamma** prior as a baseline, which can saturate in a single-sided (lower bounding variance) or double-sided (upper and lower bounding variance) manner depending on its parameters; this allows us to avoid optimization instabilities while also regularizing variance to pass our PPCs. The **Variational Posterior (VAP)** prior independently sets every local prior to its corresponding variational posterior such that the KL divergence penalty in eq. (1) vanishes. This 'prior' serves as an ablation test for the KL divergence's regularization effect. The Empirical Bayes **VAMP** prior (Tomczak & Welling, 2017) is the distribution that maximizes the ELBO: the aggregate posterior $p^*(\lambda) = N^{-1} \sum_{j=1}^{N} q(\lambda|\alpha(x_j), \beta(x_j))$, taken over the $N$ training points. For computational efficiency, Tomczak & Welling (2017) propose using $K \ll N$ randomly selected (without replacement) training points (pseudo-inputs). They denote the $j$'th pseudo-input as $u_j$ and consider optimizing pseudo-inputs $\{u_1, \ldots, u_K\}$ as trainable parameters, which we denote as **VAMP***. It is worth noting both VAMP priors are homoscedastic. For heteroscedastic priors, we first consider a novel modification to the VAMP prior, **xVAMP**, which gains heteroscedasticity by using

Table 1: Precision priors. Amortized parameters are those of shared parameter maps. We use '(\*)' to mark priors for which we tested Empirical Bayes. When '\*' appears next to a prior's name, the Empirical Bayes parameters are optimized; when absent, we fix Empirical Bayes parameters a priori.

| | | Prior Parameters | | |
|---|---|---|---|---|
| Name | Prior Form | Amortized | Empirical Bayes | Fixed |
| Gamma | $p(\lambda_i) = \Gamma(\lambda_i; a, b)$ | None | None | $a, b \in \mathbb{R}_{>0}$ |
| VAP | $p(\lambda_i|x_i) = q(\lambda_i|\alpha(x_i), \beta(x_i))$ | $\alpha(\cdot), \beta(\cdot)$ | None | None |
| VAMP(*) | $p(\lambda_i) = K^{-1} \sum_{j=1}^{K} q(\lambda_i|\alpha(u_j), \beta(u_j))$ | $\alpha(\cdot), \beta(\cdot)$ | $u_j \in \mathbb{R}^{\dim(X)}$ | $K \in \mathbb{N}_+$ |
| xVAMP(*) | $p(\lambda_i|x_i) = \sum_{j=1}^{K} \pi_j(x_i) q(\lambda_i|\alpha(u_j), \beta(u_j))$ | $\alpha(\cdot), \beta(\cdot), \pi(\cdot)$ | $u_j \in \mathbb{R}^{\dim(X)}$ | $K \in \mathbb{N}_+$ |
| VBEM(*) | $p(\lambda_i|x_i) = \sum_{j=1}^{K} \pi_j(x_i) \Gamma(\lambda_i|a_j, b_j)$ | $\pi(\cdot)$ | $a_j, b_j \in \mathbb{R}_{>0}$ | $K \in \mathbb{N}_+$ |

$\pi(\cdot)$, a neural network that maps $x_i$ onto the simplex to determine the mixture proportions. Intuitively, $\pi(x_i)$ should up weight the most relevant mixture component, whereas VAMP treats all weights uniformly. The KL divergence for xVAMP decomposes into

$$\mathbb{E}_{q(\lambda_i|x_i)}[\log q(\lambda_i|x_i)] - \mathbb{E}_{q(\lambda_i|x_i)}\left[\log \sum_{j=1}^{K} \pi_j(x_i) q(\lambda_i|u_j)\right], \tag{3}$$

where we evaluate the first term analytically as the Gamma distribution's negative entropy and Monte-Carlo (MC) estimate the second. We derive eq. (3) in our appendix. We too consider trainable pseudo-inputs for our xVAMP prior, which we denote as **xVAMP**\*. Our second heteroscedastic prior, **VBEM**, is a mixture of Gamma distributions where a trainable simplex mapping similarly determines the mixture proportions. VBEM stands for *Variational Bayes Expectation Maximization*, since optimizing the prior parameters during VI is analogous to performing M steps. VBEM's KL divergence replaces $q(\lambda|\alpha(u_j), \beta(u_j))$ with $p(\lambda|a_j, b_j)$ in eq. (3). The non-trainable set of scalar parameters $\{a_j, b_j\}_{j=1}^{K}$ is the Cartesian square of a set of scalars in $[0.05, 4.0]$ (see appendix). **VBEM**\* is the Empirical Bayes version, which randomly initializes parameters $\{\hat{a}_1, \hat{b}_1, \ldots, \hat{a}_K, \hat{b}_K\}$ from a Uniform$([-3, 3])$ and applies a softplus to ensure valid Gamma parameters (e.g. $a_j = \text{softplus}(\hat{a}_j)$). It is worth considering how variational variance might avoid zero while optimizing prior parameters: maximizing the ELBO's negative KL divergence (eq. (1)) involves maximizing the variational posterior's entropy thereby ensuring non-zero variances are integrated over in eq. (2).

## 3 Heteroscedastic Regression Experiments

In this section, we consider the task of heteroscedastic regression using Gaussian likelihoods. Section 2 details our proposal (for regression) to treat precision variationally with one of the priors from table 1, optimize the ELBO (eq. (1)), and compute the predictive distribution $p(Y_*|X_* = x_*)$ (eq. (2)). In the following paragraphs, we detail the optimization and predictive distributions of our chosen baselines and their connections to our variational treatment. Figure 3 visually compares the graphical representation of the different model families.

**The Normal model** (Nix & Weigend, 1994) parameterizes local Gaussian likelihoods with amortized mean and variance networks $p(Y_i|X_i = x_i) \triangleq \mathcal{N}(Y_i|\mu(x_i), \sigma^2(x_i))$ and performs MLE optimization by maximizing the log likelihood of the data

$$\sum_{i=1}^{N} \log \mathcal{N}(Y_i = y_i|\mu(x_i), \sigma^2(x_i)).$$

Here, $\sigma^2(\cdot)$ applies a softplus to ensure positive variances and assumes diagonal covariance. We refer to this MLE baseline as the **Normal** model. While the **Normal** model uses the same likelihood as our proposal,

MLE inference of its variance results in a predictive distribution that is simply the likelihood since there are no latent variables requiring integration. That is $p(Y_*|X_* = x_*) = \mathcal{N}(Y_*|\mu(x_*), \sigma^2(x_*))$, which is in contrast to the Student's $t$ predictictive distribution of our approach (eq. (2)). One can think of the **Normal** model as ablating our Bayesian treatment.

**The Student model** is an adaptation of the proposal of Takahashi et al. (2018) for VAEs to regression. It parameterizes local Student likelihoods with amortized mean, shape, and scale networks $p(Y_i|X_i = x_i) \triangleq \mathrm{T}(Y_i|\mu(x_i), \alpha(x_i), \beta(x_i))$ and maximizes the log likelihood of the data

$$\sum_{i=1}^{N} \log \mathrm{T}(Y_i = y_i|\mu(x_i), \alpha(x_i), \beta(x_i)) \equiv \sum_{i=1}^{N} \log \int \mathcal{N}(Y_i = y_i|\mu(x_i), \lambda_i)\Gamma(\lambda_i|\alpha(x_i), \beta(x_i))d\lambda_i.$$

Here, $\alpha(\cdot)$ and $\beta(\cdot)$ apply a softplus to ensure valid parameters. One interpretation of the **Student** model is that it performs MLE with a likelihood different from the **Normal** model and our proposal; this implies its predictive distribution is simply the new likelihood $p(Y_*|X_* = x_*) = \mathrm{T}(Y_*|\mu(x_*), \alpha(x_*), \beta(x_*))$, which gets us to the same predictive form as our proposal (eq. (2)) despite the difference in optimization. However, in this parameterization, the **Student** model integrates over Gamma distributed precisions and can thus be thought of as a Bayesian treatment of unknown precisions. In this sense, the **Student** model directly optimizes the predictive distribution of our proposal. Furthermore, the expected log likelihood of our variational objective (eq. (1)) lower bounds the **Student** model's objective. Thus, one can think of the **Student** model as ablating the variational inference aspect of our Bayesian proposal. Additionally, the modeling capacity of the **Student** model is the same as our proposal as it requires the same number of neural parameterizations. Because we use separate networks for each parameter, the computational cost of the **Student** model's three networks is simply 1.5 times the cost of the **Normal** model's two networks. Of course, one could merge multiple networks into one with a multi-headed output. If a multi-headed output occurs at the final layer, then each scalar output adds only an inner product and an activation function. We did not consider this more compact architecture in order to maintain parity with Takahashi et al. (2018); Detlefsen et al. (2019).

**The Detlefsen model** (Detlefsen et al., 2019) uses four proposals to improve predictive variance estimates, which appendix C.1 discusses in detail. We use **Detlefsen** to refer to their top method, which employs all four of their proposals. They also replace the Normal likelihood with a Gamma-Normal parameterized Student's $t$, however, they chose MC integration over analytic integration; this approximates the predictive Student, an infinite mixture of Gaussians resulting from analytic integration, with a finite mixture. The differences between the **Normal** model, the **Student** model, and our proposals thus far has been a subtle change in inference. The combination of proposals in the **Detlefsen** is largely a profound shift in inference of the **Student** model. In contrast to Detlefsen et al. (2019), we propose a single, simple modification: treat precision variationally. Furthermore, our method notably does not require enforcing that $x_i$'s nearest neighbors accompany it in a batch.

### 3.1 Toy Data

Detlefsen et al. (2019) discuss the effect neighbors have on variance estimation; despite this, their toy data process does simulate an isolated point. As such, their method appears equivalent in performance to the **Normal** model in their manuscript. We modify their toy data process to simulate heteroscedastic data with a rogue data point to better differentiate predictive mean calibration amongst the models. We sample covariates $x_i \sim \mathrm{Uniform}(0,5)$ and add $x = 7.5$ as an isolated covariate. We then generate $y \triangleq x \cdot \sin(x) + \epsilon$ where $\epsilon|x \sim \mathcal{N}(0, [0.3 \cdot (1+x)]^2)$. We consider the original process of Detlefsen et al. (2019) as well as one with fixed homoscedastic variance in appendix C.3. There, we also describe other implementation details.

In fig. 1, we compare the predictive mean and variance of our proposals (section 2), the models discussed in section 3, and a **homoscedastic Normal** model where we minimize root mean square error (RMSE) and thereafter set the variance to the mean of squared errors of the training points. We call attention to whether a model's predictive mean (the expectation of $p(Y_*|X_* = x_*)$) converges on the point at $x = 7.5$ *and* whether a model's predictive standard deviation (w.r.t. $p(Y_*|X_* = x_*)$) accurately captures the toy process's noise variance on the data-rich interval $x \in [0,5]$. We tested each model against 20 randomly sampled datasets

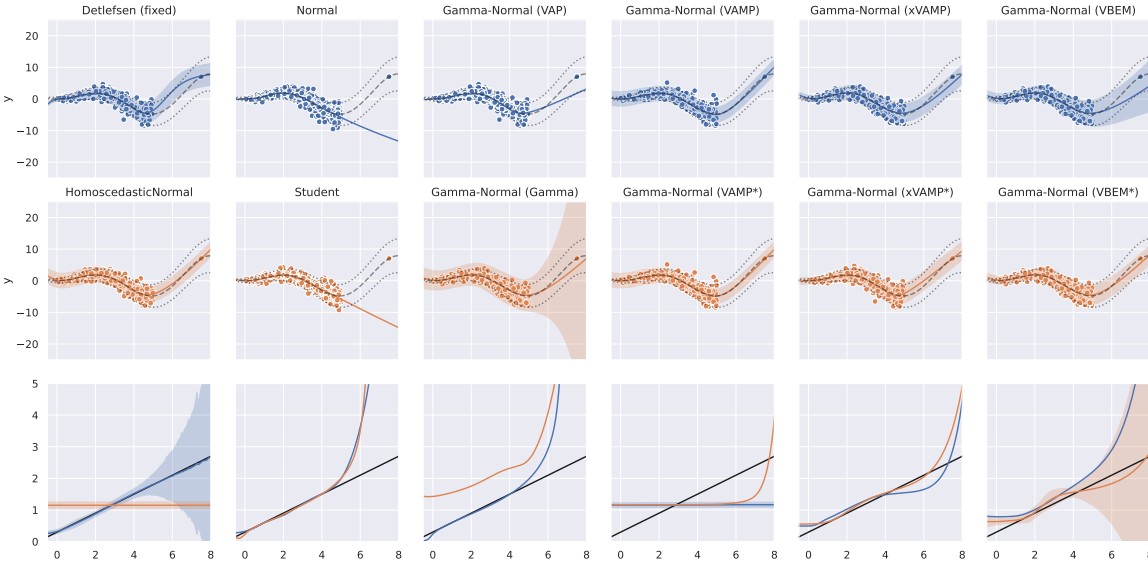

Figure 1: Toy regression results. Top two rows: dots are training data, black dashed/dotted lines and colored lines/areas are the true and predictive $\mathbb{E}[y|x] \pm 2 \cdot \sqrt{\mathrm{var}(y|x)}$, respectively. Third row: the true (black) and average predictive (colors correspond to methods above) $\sqrt{\mathrm{var}(y|x)}$ for 20 trials (area is one deviation).

and show the predictive mean calibration for the run with the best RMSE in the top two rows of fig. 1; this means that while some plots will show different data points we can be sure that models that wildly miss the point at $x = 7.5$ never reached it.

For the **Detlefsen (fixed)** model, we fixed a bug in their log likelihood code that only affected this particular experiment. Fixing it significantly improves their predictive variance–see fig. 6 in our appendix for a before-and-after of our fix. **Detlefsen (fixed)** converges on the point at $x = 7.5$ likely because of their proposal to fit mean and variance separately–the variance network is only fit after the mean network has converged–and accurately captures the underlying noise variance. The **homoscedastic Normal** model converges on the point at $x = 7.5$, which suggests the mean network is sufficiently flexible to reach it. Of course, the global homoscedastic variance can only capture the mean of true noise variance. In contrast, the **Normal** and **Student** models accurately capture noise variance for $x \in [0, 5]$ but fail to converge on the point at $x = 7.5$. Given the mean network was able to converge for models that fit it in isolation (e.g. **homoscedastic Normal** and **Detlefsen (fixed)**), we can be relatively certain that the **Normal** and **Student** models mean networks fail to converge due to the fact they jointly optimize mean and variance. Furthermore, the **Normal** and **Student** models can converge on the point at $x = 7.5$ when data is added for $x \in [5, 10]$ (see the appendix C.3 and fig. 4).

With regard to our proposals, the VAP prior was the only prior that missed the point at $x = 7.5$ by more than two true standard deviations. *All* other considered priors (i.e. those that do not ablate the regulatory effect of precision's KL divergence) fell within two true standard deviations and often much closer; this largely confirms the beneficial regulatory effect of our variational treatment of precision. The Gamma prior seemingly provides too much regularization given its overestimation of predictive variance for $x \in [0, 5]$; we suspect additional tuning would improve this. The VAMP and VAMP* priors are poor at capturing heteroscedastic variance because $D_{KL}(q(\lambda|x) || K^{-1} \sum_{j=1}^{K} q(\lambda|u_j))$ is minimized when the variational distributions are uniform. Indeed, they predict homoscedasticity with a nearly constant standard deviation that is approximately equal to the expected value of the true standard deviation over the training interval. Our xVAMP(*) and VBEM(*) exhibit well calibrated predictive uncertainty across the test interval $[-0.5, 8]$.

### 3.2 UCI Data

We consider many of the same UCI datasets as Detlefsen et al. (2019). We independently normalize covariates and targets to zero mean and unit variance, but report metrics for the original target scalings. Remaining implementation specifics match Detlefsen et al. (2019) (see appendix). We perform the following PPCs on randomly held-out validation sets that each constitute 10% of the data across 20 trials:

- average log predictive likelihood: $|\mathcal{D}_{\text{test}}|^{-1} \sum\limits_{(x_*,y_*)\in\mathcal{D}_{\text{test}}} \log p(Y_* = y_* | X_* = x_*)$

- RMSE of the predictive mean: $\left( |\mathcal{D}_{\text{test}}|^{-1} \sum\limits_{(x_*,y_*)\in\mathcal{D}_{\text{test}}} \left( \mathbb{E}[Y_*|X_* = x_*] - y_* \right)^2 \right)^{\frac{1}{2}}$

- bias of the predictive variance $|\mathcal{D}_{\text{test}}|^{-1} \sum\limits_{(x_*,y_*)\in\mathcal{D}_{\text{test}}} \text{var}[Y_*|X_* = x_*] - \left( \mathbb{E}[Y_*|X_* = x_*] - y_* \right)^2$

- RMSE of a predictive sample: $\left( |\mathcal{D}_{\text{test}}|^{-1} \sum\limits_{(x_*,y_*)\in\mathcal{D}_{\text{test}}} \left( \text{sample from } p(Y_*|X_* = x_*) - y_* \right)^2 \right)^{\frac{1}{2}}$

Recall from sections 2 and 3, $p(Y_*|X_* = x_*)$ is a Student's $t$ for all methods except the Normal model, which is Normal. All expectations and variances are w.r.t. the predictive distribution $p(Y_*|X_* = x_*)$, which has mean $\mu(x_*)$ for all models. The Normal model's predictive variance is $\sigma^2(x_*)$ (i.e. the output of the variance network). For the remaining models with a Student's $t$ predictive distribution, the predictive variance is $\forall \alpha(x_*) > 1 : \frac{\beta(x_*)}{\alpha(x_*)-1}$ (i.e. the expectation of an Inverse-Gamma), which is always available since we offset $\alpha(\cdot)$'s softplus output by 1. We emphasize this adjustment still allows variances arbitrarily close to zero and infinity. For variance, we compute bias to understand when predictive uncertainty is over/under estimated.

We jointly report log predictive likelihood and predictive variance bias in table 2. We include independent tables for each of our four proposed PPCs in our appendix alongside recent log likelihoods and mean RMSEs (Sun et al., 2019; Sicking et al., 2021), which we generally match. We tally the number of datasets for which a method was the top PPC performer or was statistically indistinguishable from the winner according to a two-sided Kolmogorov–Smirnov test with $p \leq 0.05$ (appendix table 4). VBEM* offers the best log predictive likelihood and is the overall top performer. Detlefsen has the lowest predictive likelihoods (table 2), even against our implementation of the Normal baseline. Their code did not support multivariate $y$–we modified it to do so and remain unsure how they generated their results for UCI data with multidimensional targets. Perhaps, they labeled covariates and targets differently and/or performed additional preprocessing (their code loads unprovided numpy files as its data source). For naval, rescaling their algorithm's MC-sampled variances to the original target scalings produces very small values that introduce NaNs when estimating the variance of their predictive Student via a mixture of Gaussians parameterized by these small variances. The Normal, Student, and VAP baselines exhibit severely biased and wildly varying predictive variance on about half the datasets compared to our other methods, reaffirming the benefit of our probabilistic regularization of variance. The Student baseline achieves top likelihood on both carbon and power plant, for which its predictive variance calibration ranges from very poor to excellent; this paradox highlights the need for looking beyond just likelihood. The VAMP(*) and VBEM* *never* exhibit wildly varying predictive variance and generally produce top-performing estimates. xVAMP(*) exhibits poor variance calibration only on yacht. We repeat Detlefsen et al. (2019)'s active learning experiments (see appendix) and find VBEM* is most frequently the top performer for these same UCI datasets.

## 4 Variational Autoencoder Experiments

In this section, we consider VAEs that use per-pixel heteroscedastic Gaussian likelihoods. To the best of our knowledge, Takahashi et al. (2018); Detlefsen et al. (2019) are the only other manuscripts that have considered improving the optimization and performance for this class of model. VAE papers using Gaussian likelihoods that claim improvements to sample quality (van den Oord et al., 2017; Razavi et al., 2019) and imputation (Nazabal et al., 2018; Mattei & Frellsen, 2018b) do not test per-pixel heteroscedastic Gaussian

Table 2: UCI log predictive likelihood and predictive variance bias (mean±std.). Tuples appearing below dataset are $(N_{\text{observations}}, \dim(x), \dim(y))$. Winners are in bold, statistical ties are not.

| Algorithm | Prior | boston (506, 13, 1) | carbon (10721, 5, 3) | concrete (1030, 8, 1) | energy (768, 8, 2) | naval (11934, 16, 2) |
|---|---|---|---|---|---|---|
| | | Log Predictive Likelihood | | | | |
| Detlefsen | N/A | -2.98±0.09 | 8.77±0.24 | -3.66±0.08 | -4.90±0.27 | 9.67±0.19 |
| Normal | N/A | -2.42±0.23 | 13.20±1.35 | -3.06±0.17 | -0.48±0.69 | 14.15±0.17 |
| Student | N/A | -2.37±0.19 | **17.19±0.21** | -3.10±0.17 | 0.22±0.31 | 13.60±0.39 |
| Gamma-Normal | VAP | -2.36±0.17 | 15.52±0.24 | -3.12±0.17 | 0.17±0.44 | 13.36±0.41 |
| | Gamma | -2.48±0.29 | 11.28±0.02 | -3.20±0.16 | -1.05±0.18 | 12.33±0.16 |
| | VAMP | -2.39±0.17 | 14.37±0.17 | -3.09±0.16 | -0.18±0.21 | 14.16±0.78 |
| | VAMP* | -2.39±0.16 | 14.38±0.12 | -3.09±0.16 | -0.16±0.20 | 13.96±0.88 |
| | xVAMP | **-2.33±0.17** | 15.38±0.24 | -3.01±0.14 | 0.05±0.28 | 13.50±0.59 |
| | xVAMP* | -2.33±0.17 | 15.41±0.18 | -3.01±0.13 | 0.11±0.39 | 13.34±0.47 |
| | VBEM | -2.46±0.11 | 4.57±1.00 | -3.11±0.07 | -4.52±0.26 | 9.02±0.61 |
| | VBEM* | -2.36±0.14 | 14.64±0.16 | **-2.99±0.13** | **0.49±0.28** | **14.42±0.15** |
| | | Predictive Variance Bias | | | | |
| Detlefsen | N/A | 1.0e+02±79.11 | 9.8e-05±1.6e-04 | 2.2e+02±91.85 | 18.60±8.88 | nan±nan |
| Normal | N/A | 31.63±1.5e+02 | 3.5e+23±1.6e+24 | -2.01±8.67 | -0.16±0.24 | 3.1e-07±2.0e-06 |
| Student | N/A | 18.08±63.79 | 0.12±0.23 | -2.20±9.28 | 24.00±85.42 | 4.9e-06±2.2e-05 |
| Gamma-Normal | VAP | 3.3e+02±1.3e+03 | 0.25±1.11 | -2.13±7.85 | 0.04±0.31 | 3.1e-07±6.8e-07 |
| | Gamma | 3.18±20.09 | 1.5e-04±5.8e-05 | 0.76±11.35 | 0.31±0.40 | 3.7e-06±2.9e-06 |
| | VAMP | -2.96±7.96 | -6.6e-06±6.0e-05 | -6.15±5.18 | -0.15±0.39 | 1.7e-07±2.9e-07 |
| | VAMP* | -3.00±7.84 | **-6.0e-06±6.0e-05** | -6.16±5.18 | -0.13±0.40 | **1.3e-07±3.3e-07** |
| | xVAMP | 0.65±16.20 | 2.7e-05±9.7e-05 | -4.82±4.63 | **6.0e-03±0.36** | 3.1e-07±7.4e-07 |
| | xVAMP* | 0.51±20.17 | 5.0e-04±2.2e-03 | -4.66±5.07 | -8.5e-03±0.36 | 2.7e-07±6.3e-07 |
| | VBEM | 6.74±8.48 | 0.01±4.5e-03 | 25.86±8.94 | 22.06±5.58 | 3.6e-05±1.4e-05 |
| | VBEM* | **-0.11±8.62** | -7.2e-06±6.1e-05 | **-0.58±5.05** | 0.02±0.28 | 3.9e-07±4.9e-07 |

| Algorithm | Prior | power plant (9568, 4, 1) | superconductivity (21263, 81, 1) | wine-red (1599, 11, 1) | wine-white (4898, 11, 1) | yacht (308, 6, 1) |
|---|---|---|---|---|---|---|
| | | Log Predictive Likelihood | | | | |
| Detlefsen | N/A | -3.26±9.1e-03 | -5.21±0.02 | -1.04±0.06 | -1.12±0.04 | -3.15±0.10 |
| Normal | N/A | -2.82±0.05 | -3.51±0.10 | -0.92±0.05 | -1.05±0.04 | -1.55±0.65 |
| Student | N/A | **-2.78±0.03** | -3.41±0.05 | **-0.80±0.10** | -1.05±0.04 | -1.73±0.59 |
| Gamma-Normal | VAP | -2.81±0.04 | -3.45±0.06 | -0.87±0.06 | -1.04±0.04 | -1.79±0.50 |
| | Gamma | -2.88±0.03 | -3.45±0.04 | -0.98±0.07 | -1.13±0.05 | -1.73±0.38 |
| | VAMP | -2.83±0.03 | -3.94±0.02 | -0.94±0.05 | -1.05±0.04 | -2.83±0.70 |
| | VAMP* | -2.83±0.03 | -3.94±0.03 | -0.94±0.05 | -1.05±0.04 | -2.77±0.77 |
| | xVAMP | -2.81±0.04 | -3.40±0.04 | -0.90±0.05 | -1.03±0.04 | -1.68±0.38 |
| | xVAMP* | -2.81±0.04 | **-3.39±0.05** | -0.89±0.06 | -1.03±0.04 | -1.71±0.47 |
| | VBEM | -2.89±0.05 | -3.77±0.09 | -0.91±0.05 | -1.03±0.03 | -2.64±0.23 |
| | VBEM* | -2.81±0.03 | -3.41±0.04 | -0.89±0.06 | **-1.03±0.04** | **-1.11±0.57** |
| | | Predictive Variance Bias | | | | |
| Detlefsen | N/A | 69.25±2.40 | 5.5e+04±6.2e+03 | 2.16±1.57 | 0.83±0.36 | 96.62±54.08 |
| Normal | N/A | **0.05±1.53** | 2.3e+13±1.0e+14 | -3.8e-03±0.04 | -0.02±0.06 | 20.68±54.95 |
| Student | N/A | -0.27±1.47 | 1.6e+05±3.3e+05 | 12.52±30.71 | -5.6e-03±0.05 | 1.7e+03±2.3e+03 |
| Gamma-Normal | VAP | 0.52±1.29 | 9.0e+05±2.6e+06 | 0.03±0.05 | 0.13±0.64 | 1.3e+03±1.5e+03 |
| | Gamma | 2.34±1.43 | 1.1e+02±81.21 | 0.04±0.11 | **-2.3e-03±0.05** | **-7.28±40.88** |
| | VAMP | 0.89±1.04 | **-9.83±7.97** | 0.05±0.06 | -8.8e-03±0.04 | 38.05±83.39 |
| | VAMP* | 0.89±1.04 | -9.89±7.98 | 0.05±0.06 | -8.9e-03±0.04 | 38.07±83.29 |
| | xVAMP | 0.46±1.25 | 14.40±42.90 | 3.5e-03±0.05 | -0.03±0.03 | 4.8e+02±1.7e+03 |
| | xVAMP* | 0.44±1.24 | 1.3e+02±4.7e+02 | **2.1e-03±0.05** | -0.03±0.03 | 1.7e+02±1.5e+02 |
| | VBEM | 16.53±9.32 | 91.44±25.39 | 0.08±0.04 | 0.07±0.04 | 20.70±25.23 |
| | VBEM* | 1.86±1.44 | 9.87±16.22 | 0.05±0.06 | 0.01±0.04 | 26.48±26.88 |

likelihoods and do not sample the predictive distribution despite sometimes fitting a global (homoscedastic) scalar noise variance to improve mean calibration (Dai & Wipf, 2019). These methods ancestrally resample latent variables from the variational posterior (or prior) and report the expected value of the decoder density. This procedure is actually a Monte-Carlo estimate of the reconstruction mean $\mathbb{E}[X_*|X = x]$ (or prior predictive mean $\mathbb{E}[X_*]$). Reporting the expectation as a sample obfuscates predictive uncertainty. More recently, Vahdat & Kautz (2020) employ per-pixel heteroscedastic scale parameters but with a discretized logistic mixture likelihood (Salimans et al., 2017) to achieve state-of-the-art VAE image sampling. Their code employs a clamp that prevents their scale parameter from approaching zero suggesting brittle optimization may still exist. Adapting our proposals to this alternative likelihood is compelling, but beyond the scope of this work. Following the same the flow in section 3, the following paragraphs describe the optimization and

predictive distributions of the VAE models we consider. Figure 9 in our appendix compares the graphical models for the VAE variants discussed below.

**The VAE** (Kingma & Welling, 2013; Rezende et al., 2014) is a deep latent variable model that provides computationally efficient VI for a generative process from a low-dimensional latent local Gaussian variable $Z$ to high-dimensional data $X$. Any (decoder) distribution can be placed over $X$, but we specifically focus on Gaussian likelihoods $p(X|Z = z) \triangleq \mathcal{N}(X|\mu_x(z), \sigma_x^2(z))$. As is common, we use $p(Z) \triangleq \mathcal{N}(0, I)$ as the prior, posit $q(Z|X = x) \triangleq \mathcal{N}(Z|\mu_z(x), \sigma_z^2(x))$ as the variational family (encoding distribution), and perform black-box VI (Ranganath et al., 2014) with reparameterization gradients (Salimans et al., 2013; Figurnov et al., 2018) to maximize the ELBO,

$$\sum_{x \in \mathcal{D}} \mathbb{E}_{q(z|x)} \Big[ \log \mathcal{N}(x|\mu_x(z), \sigma_x^2(z)) \Big] - D_{KL}\big(q(z|x) \,||\, p(z)\big). \tag{4}$$

Encoder maps, $\mu_z(x)$ and $\sigma_z^2(x)$, are bifurcated outputs of the same neural network as is common practice. We explore decoder parameter maps $\mu_x(z)$ and $\sigma_x^2(z)$ as being either bifurcated outputs of the same neural network (**VAE**) or separate neural networks (**VAE-Split**). Additionally, we evaluate batch normalization's effect ($+$ **BN**). Softplus activations ensure positive variance.

Defining a posterior predictive for VAEs is nuanced and unfortunately rarely discussed in the literature. Alemi et al. (2018) refer to $p(X_*) = \mathbb{E}_{\hat{p}(X)} \mathbb{E}_{q(Z|X)}[p(X_*|Z)]$ as the *empirical data reconstruction distribution*, where $\hat{p}(X)$ approximates the true data generating distribution by uniformly sampling from a sufficiently large dataset. Computing $p(X_*)$'s outer expectation for $N$ data points and approximating the inner expectation with $M$ Monte-Carlo samples requires evaluating $MN$ mixture components. Conditioning just the inner expectation on a specific $x$ yields what we call the *local reconstruction distribution*, $p(X_*|X = x) = \mathbb{E}_{q(Z|X=x)}[p(X_*|Z)]$, a more manageable $M$-component mixture. This distribution implicitly conditions on $\mathcal{D}$, $p(X_*|X = x, \mathcal{D})$, since parameters maps for $p(X_*|Z)$ and $q(Z|X)$ were fit using $\mathcal{D}$. Calling $p(X_*|X = x)$ the *variational posterior predictive* distribution is a slight abuse of Bayesian lexicon, yet we do so to maintain harmony with our regression methods. We Monte-Carlo estimate $p(X_*|X = x) = \mathbb{E}_{q(z|x)}[\mathcal{N}(X_*|\mu_x(z), \sigma_x^2(z))]$ with 20 $z$ samples, which yields a uniform mixture of 20 Gaussians.

**The Student VAE** (Takahashi et al., 2018) uses a Student's $t$ likelihood and exhibits improved optimization stability and predictive likelihood, but has not had its predictive mean and variance calibration examined. Their ELBO is similar to eq. (4) but with a Student's $t$ likelihood $\mathrm{T}(X|\mu_x(z), \lambda_x(z), \nu_x(z))$ parameterized by three separate neural networks, $\mu_x(z)$, $\lambda_x(z)$ and $\nu_x(z)$ for mean, precision, and degrees-of-freedom, respectively. Since the Student's $t$ variance is undefined for $\nu_x(z) \in (0, 1]$, infinite for $\nu_x(z) \in (1, 2]$, and arbitrarily close to $\infty$ for $\nu_x(z) \approx 2$, we restrict $\nu_x(z) > 3$ using a shifted softplus. We found that allowing the posterior predictive to attain these high variances worsens its PPC performance beyond what we report. Having replaced the Normal likelihood with a Student's $t$, it follows that the Student VAE's posterior predictive is $p(X_*|X = x) = \mathbb{E}_{q(z|x)}[\mathrm{T}(X|\mu_x(z), \lambda_x(z), \nu_x(z))]$, which we again approximate with 20 $z$ samples yielding uniform mixture of 20 Student's $t$. Takahashi et al. (2018) also propose the **MAP-VAE**, where precision is absorbed into the likelihood: $p(\lambda|z) \triangleq \mathrm{Gamma}(\lambda_x(z); a, b)$ for pre-defined constants $a$ and $b$. The **MAP-VAE**'s ELBO adds $\mathbb{E}_{q(z|x)}[\log p(\lambda|z)]$ to eq. (4) and has the same $p(X_*|X = x)$ as the VAE.

**The Detlefsen VAE** (Detlefsen et al., 2019) adapts their regression proposals to VAEs. We refer the reader to their manuscript for additional details.

**V3AE** (variational variance VAE) treats precision variationally with $q(\lambda|z) \triangleq \mathrm{Gamma}(\lambda|\alpha(z), \beta(z))$. We use the priors discussed in section 2, except we now condition on latent codes $z_i$. The resulting ELBO for V3AE,

$$\sum_{x \in \mathcal{D}} \mathbb{E}_{q(z|x)} \left[ \mathbb{E}_{q(\lambda|z)} \Big[ \log \mathcal{N}(x|\mu_x(z), \lambda) \Big] - D_{KL}\big(q(\lambda|\alpha(z), \beta(z)) \,||\, p(\lambda)\big) \right] - D_{KL}\big(q(z|x) \,||\, p(z)\big),$$

introduces a KL divergence that regularizes the predictive variance. This proposal with an appropriate prior addresses the theoretical preference of an optimal decoder for zero variance (Dai & Wipf, 2019) and the theoretical concern that continuous VAEs are ill-posed with unbounded likelihood functions (Mattei & Frellsen, 2018a). Our V3AE has two variational distributions, $q(z|x)$ and $q(\lambda|z)$, such that the predictive distribution $p(X_*|X = x) = \mathbb{E}_{q(z|x)q(\lambda|z)}[\mathcal{N}(X_*|\mu_x(z), \lambda)]$. We integrate V3AE's normal likelihood w.r.t. $q(\lambda|z)$ analytically, yielding a Student's $t$ (as in eq. (2)). Thereafter, MC integration w.r.t. $q(z|x)$ yields a uniform mixture of 20 Student's $t$. See appendix for additional details (e.g. network architectures).

## 4.1 VAE Results

For VAEs, there is a subtle distinction between the *expected log likelihood* $\mathbb{E}_{q(Z|X=x)}[\log p(X_* = x|Z)]$ from the variational objective and the *log reconstruction likelihood* $\log \mathbb{E}_{q(Z|X=x)}[p(X_* = x|Z)]$ and authors rarely clarify which "log likelihood" they report. We use local reconstruction distribution $p(X_*|X = x)$ as the posterior predictive distribution for our PPCs. Similar to regression, we consider the following PPCs:

- average log predictive likelihood: $|\mathcal{D}_{\text{test}}|^{-1} \sum_{x \sim \mathcal{D}_{\text{test}}} \log p(X_* = x|X = x)$

- RMSE of the predictive mean: $\left(|\mathcal{D}_{\text{test}}|^{-1} \sum_{x \sim \mathcal{D}_{\text{test}}} \left(\mathbb{E}[X_*|X = x] - x\right)^2\right)^{\frac{1}{2}}$

- bias of the predictive variance $|\mathcal{D}_{\text{test}}|^{-1} \sum_{x \sim \mathcal{D}_{\text{test}}} \text{var}[X_*|X = x] - \left(\mathbb{E}[X_*|X = x] - x\right)^2$

- RMSE of a predictive sample: $\left(|\mathcal{D}_{\text{test}}|^{-1} \sum_{x \sim \mathcal{D}_{\text{test}}} \left(\text{sample from } p(X_*|X = x) - x\right)^2\right)^{\frac{1}{2}}$

Except for the first PPC, the above notation intentionally ignores the fact that $x$ is a high-dimensional vector in an effort to simplify notation. Specifically, we apply these PPCs on a per-pixel basis and take the average over all pixes. Comparing ELBOs (including tighter importance weighted ELBOs (Burda et al., 2015)) across these methods is problematic. The marginal likelihood $p(x)$ is $\mathbb{E}_{p(z)}[\mathcal{N}(x|\mu_x(z), \sigma_x^2(z))]$ for VAE(-Split)(+BN), $\mathbb{E}_{p(z)}[\text{T}(x|\mu_x(z), \lambda_x(z), \nu_x(z))]$ for VAE-Student, and $\mathbb{E}_{p(z)p(\lambda)}[\mathcal{N}(x|\mu_x(z), \lambda)]$ for V3AE. Hence, each parameterizations has a different $p(x)$. Comparing lower bounds for different $p(x)$ is meaningless, unfortunately.

Table 3: VAE PPCs for Fashion MNIST (mean±std.)

| Method | LL | Mean RMSE | Var Bias | Sample RMSE |
|---|---|---|---|---|
| Fixed-Var. VAE (1.0) | -730.05±0.11 | 0.15±1.8e-03 | 0.98±5.7e-04 | 1.01±3.7e-04 |
| Fixed-Var. VAE (0.001) | -1452.44±3.65 | **9.4e-02±4.6e-05** | -7.8e-03±8.6e-06 | **9.9e-02±4.7e-05** |
| VAE | 2154.31±42.11 | 0.25±1.4e-03 | 3.3e-02±1.5e-03 | 0.39±3.1e-03 |
| VAE + BN | 1639.39±15.33 | 0.20±1.8e-03 | 2.1e-02±3.0e-03 | 0.31±5.6e-03 |
| VAE-Split | 2099.28±39.97 | 0.27±2.9e-03 | 4.7e-02±1.6e-03 | 0.45±4.8e-03 |
| VAE-Split + BN | 1948.30±25.87 | 0.26±2.6e-03 | 3.1e-02±3.6e-03 | 0.41±1.1e-02 |
| Detlefsen (0.001) | -7214.05±55.55 | 0.15±4.7e-04 | -2.1e-02±1.1e-03 | 0.16±4.1e-03 |
| Detlefsen (0.25) | -213.89±0.12 | 0.15±2.6e-04 | 0.23±7.9e-05 | 0.52±1.1e-04 |
| Detlefsen (10.0) | -1623.98±5.1e-03 | 0.15±6.2e-04 | 9.98±2.5e-04 | 3.17±5.3e-04 |
| MAP-VAE | 1003.51±32.75 | 0.11±6.8e-04 | -9.1e-03±6.2e-04 | 0.13±4.8e-03 |
| Student-VAE | **3134.52±18.60** | 0.29±3.3e-03 | 7.4e-02±1.6e-02 | 0.49±2.2e-02 |
| V3AE-VAP | 2146.46±67.83 | 0.28±3.5e-03 | **9.9e-04±3.0e-03** | 0.40±8.2e-03 |
| V3AE-Gamma | 1201.95±25.25 | 0.11±2.8e-03 | -8.0e-03±4.1e-04 | 0.12±3.4e-03 |
| V3AE-VAMP | 1632.22±12.89 | 0.17±1.3e-03 | 1.5e-03±2.7e-04 | 0.25±2.4e-03 |
| V3AE-VAMP* | 1630.10±17.87 | 0.18±2.6e-03 | 1.3e-03±2.5e-04 | 0.25±3.4e-03 |
| V3AE-xVAMP | 1601.60±21.49 | 0.18±2.5e-03 | 1.3e-03±4.1e-04 | 0.25±3.8e-03 |
| V3AE-xVAMP* | 1619.97±25.95 | 0.18±3.3e-03 | 1.5e-03±4.6e-04 | 0.25±5.0e-03 |
| V3AE-VBEM | 306.46±1.04 | 0.10±6.8e-04 | 6.4e-02±9.7e-05 | 0.29±3.3e-04 |
| V3AE-VBEM* | 1153.11±4.20 | 0.10±5.5e-04 | **4.4e-04±3.9e-05** | 0.15±8.0e-04 |

In fig. 2, we curate a subset of the VAE methods to qualitatively visualize our PPCs for MNIST and a downsampled Celeb-a. We include additional PPC visualizations for all methods in our appendix. Table 3

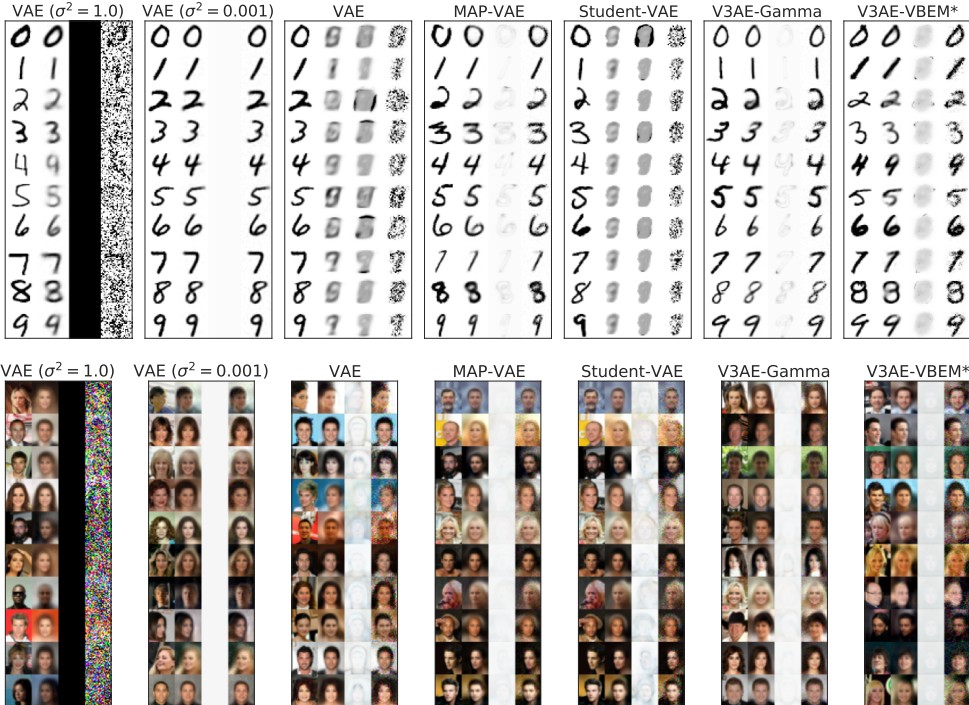

Figure 2: VAE PPCs for MNIST and downsampled Celeb-a: Columns of a subplot, left to right, are randomly selected test data followed by the predictive mean and variance and a sample from the predictive distribution. We clamp pixel values to $[0, 1]$ and invert RGB variances.

bolds top performers and statistical ties (using the same test from section 3.2) for all four of our PPCs on Fashion MNIST. We include tabular results for MNIST and Celeb-a in our appendix. We include baselines with a fixed global scalar variance to confirm variance impacts mean quality (Dai & Wipf, 2019). For MNIST and Fashion MNIST, the predictive mean estimates are better calibrated when fixing variance to 0.001 than to 1.0; this confirms the mean network has the flexibility to produce well-calibrated predictive mean estimates. However, we see no such difference for Celeb-a; this occurs since the predictive task has grown in complexity as we move from greyscale to RGB and consider non-homogeneous backgrounds. Here, the added data complexity prohibits the same-sized network from collapsing to Dirac densities–it is not flexible enough to do so–and thus all methods produce similar quality predictive mean estimates that represent the flexibility limit of the network. Returning to the problematic MNIST and Fashion MNIST data, the Normal and Student models, like their regression counterparts, exhibit poor predictive mean and variance estimates, which hide behind high model likelihoods (table 3). Figure 2 captures these models using high variance to explain MNIST data. The Detlefsen VAE employs nearest neighbors to estimate heteroscedastic variance and then extrapolates between this estimate and some fixed value, which we varied (see parentheses in table 3). For each value, we observed this mechanism latching onto the fixed value for most data points suggesting their methods do not generalize to high-dimensional data. The MAP-VAE and our V3AE-Gamma make up much of likelihood lost when fixing variance to 0.001 and exhibit well calibrated predictive mean and variance enabling predictive samples that resemble the data. Again, VBEM* generally does well in all PPC categories. Perhaps intuitively, the methods with well-calibrated predictive variances indicate the model struggles most at edge localization. Conversely, the baseline methods' predictive uncertainty is largely uninterpretable.

## 5 Conclusion

This article addresses poor predictive mean and variance calibrations resulting from MLE optimization of heteroscedastic Gaussian likelihoods that employ amortized neural-network parameter maps. When a mean

network possesses sufficient flexibility to place Dirac densities on the targets, unbounded gradients result in poorly calibrated mean estimates that can be explained away with high noise variance estimates. Thus, we caution against selecting models on the basis of log likelihood alone. Data lacking meaningful nearby neighbors provide this flexibility, which motivates existing solutions that ensure every batch member is accompanied by its nearest neighbors (Detlefsen et al., 2019). Our attractively simple solution, to treat noise variance variationally with an amortized variational family, preserves heteroscedasticity in the predictive distribution and provides a probabilistically principled method to regularize optimization away from destabilizing Dirac densities. Notably, our solution does not require finding stabilizing neighbors for every data point, which are not guaranteed to exist and requires defining a distance metric in the input space. We find that solutions relying on nearest neighbors break down in high dimensions, which is consistent with the theory that nearest neighbor distances become meaningless in high dimensions. We propose PPCs to measure predictive mean and variance calibration and find that our claims are empirically supported. Our variational Empirical Bayes methods coupled with our novel priors, particularly VBEM*, provide substantial and tangible improvements to predictive mean and variance calibration on a variety of tasks.

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

# A   Appendix

Hereafter, we include supplementary material for our manuscript. Our code is available as part of this submission. We organize this appendix using the same (major) section names as the main article. Any reference to the appendix from the main article will appear in the corresponding (major) section. Figure and table numbers continue from the main article.

# B   Variational Empirical Bayes for Noise Variance

Variational variance in regression is not novel by itself. Menictas & Wand (2015) propose CAVI (coordinate ascent variational inference) to speed up inference over MCMC methods in a fully Bayesian treatment of heteroscedastic spline regression. CAVI employs closed-form updates that provably increase the ELBO monotonically (i.e. no optimization instabilities), rather than (stochastic) gradient steps. Thus, we feel well distinguished from Menictas & Wand (2015)–we improve mean and variance calibration of a different model with a different (arguably more general) type of variational inference.

## B.1   Analytic Integration of ELBO's Expected Log Likelihood

The first expectation of eq. (1) (main article) evaluates analytically as

$$\frac{1}{2}\Big(\psi(\alpha(x)) - \log\beta(x) - \log(2\pi) - \frac{\alpha(x)}{\beta(x)}(y - \mu(x))^2\Big) \tag{5}$$

($\psi(\cdot)$ is the Digamma function) for univariate $y$ and, with a diagonal covariance assumption, for multivariate $y$.

## B.2   Derivation of xVAMP ELBO

We derive the xVAMP ELBO and decompose its KL divergence. The xVAMP generative process is

$$u_1, \ldots, u_K \sim \text{UniformWithoutReplacement}(\{x_1, \ldots, x_N\})$$

$$\lambda_i | x_i \sim p(\lambda_i | x_i, u_1, \ldots, u_K) \triangleq \sum_{j=1}^{K} \pi_j(x_i) \cdot q(\lambda_i | u_j)$$

$$y_i | x_i, \lambda_i \sim p(y_i | x_i, \lambda_i) \triangleq \mathcal{N}(y_i | \mu(x_i), \lambda_i).$$

We treat $\{u_j\}_{j=1}^{K}$ as prior parameters (not random variables). The resulting local (per-point) ELBO is

$$
\begin{aligned}
\log p(y_i|x_i) &= \underset{q(\lambda_i|x_i)}{\mathbb{E}} \left[ \log p(y_i|x_i, \lambda_i) - \log\frac{q(\lambda_i|x_i)}{p(\lambda_i|x_i)} + \log\frac{q(\lambda_i|x_i)}{p(\lambda_i|x_i, y_i)} \right] \\
&= \underset{q(\lambda_i|x_i)}{\mathbb{E}} \left[ \log p(y_i|x_i, \lambda_i) \right] - D_{KL}\big(q(\lambda_i|x_i)||p(\lambda_i|x_i)\big) + D_{KL}\big(q(\lambda_i|x_i)||p(\lambda_i|x_i, y_i)\big) \\
&\geq \underset{q(\lambda_i|x_i)}{\mathbb{E}} \left[ \log p(y_i|x_i, \lambda_i) \right] - D_{KL}\big(q(\lambda_i|x_i)||p(\lambda_i|x_i)\big) \\
&= \underset{q(\lambda_i|x_i)}{\mathbb{E}} \left[ \log p(y_i|x_i, \lambda_i) - \log q(\lambda_i|x_i) + \log\sum_{j=1}^{K} \pi_j(x_i)q(\lambda_i|u_j) \right].
\end{aligned}
$$

From the ELBO, we determine

$$
\begin{aligned}
D_{KL}\big(q(\lambda_i|x_i)||p(\lambda_i|x_i)\big) &= \underset{q(\lambda_i|x_i)}{\mathbb{E}} \left[ \log q(\lambda_i|x_i) - \log\sum_{j=1}^{K} \pi_j(x_i)q(\lambda_i|u_j) \right] \\
&= -\mathbb{H}\big[q(\lambda_i|x_i)\big] - \underset{q(\lambda_i|x_i)}{\mathbb{E}} \left[ \log\sum_{j=1}^{K} \pi_j(x_i)q(\lambda_i|u_j) \right]. \tag{6}
\end{aligned}
$$

### B.3 VBEM Parameter Set

For VBEM's prior parameters we use the Cartesian square of a set of scalars ranging from 0.05 to 4.0. That set of integers is

$$\{0.05, 0.1, 0.25, 0.5, 0.75, 1.0, 1.5, 2.0, 2.5, 3.0, 3.5, 4.0\}.$$

## C  Heteroscedastic Regression Experiments

Figure 3 contains the graphical models we refer to from the main report. In the rightmost model, where we treat local precision variationally, one can draw an arrow from $x_i$ to $\lambda_i$ without introducing a cycle in the generative process, confirming the validity of our heteroscedastic priors $p(\lambda_i|x_i)$. As depicted, the model uses a homoscedastic prior $p(\lambda_i)$. We are not generative w.r.t. $x$, but one could model $x$ and maintain validity so long as no generative cycles exist.

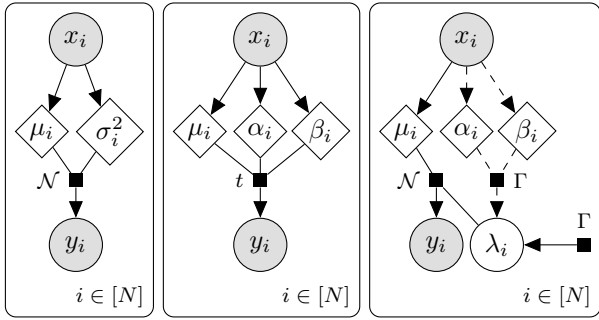

Figure 3: Graphical models for regression: Normal, Student's $t$, and Variational Variance (left to right). Diamonds are deterministic neural network parameter maps. Solid arrows denote the generative process. Dashed arrows define the variational family.

### C.1  Discussion of Detlefsen et al. (2019)

Detlefsen et al. (2019) argue a batch containing $x_i$, but lacking other nearby data, while sufficient for updating the mean, is insufficient for updating the variance. Accordingly, they propose a 'locality sampler' that ensures any batch sample $(x_i, y_i)$ is accompanied by its $K$ nearest neighbors (w.r.t. $x_i$). Unfortunately, nearest-neighbor distance can produce meaningless relationships in high dimensions. Second, they optimize the mean and variance networks in isolation. The first half of training fits only the mean network (using a fixed variance) to ensure that, during the latter half of training where coordinate ascent alternates every few batches, variance estimation is feasible since the mean network is presumably now reasonable. Gradient-based coordinate ascent complicates optimization and may introduce interplay between the two separate adaptive gradient optimizers. Third, they replace the Normal likelihood with a Gamma-Normal parameterized Student's $t$, $\mathrm{T}(y_i|x_i) \equiv \int_0^\infty \mathcal{N}(y_i|\mu(x_i), \lambda_i) \, \mathrm{Gamma}(\lambda_i|\alpha(x_i), \beta(x_i)) d\lambda_i$, which they Monte-Carlo integrate. Student's $t$ variance can be undefined and arbitrarily close to $\infty$, which makes it famously robust against outliers, but unfortunately can hamstring its ability to generate sensible data under our PPC framework. Lastly, they extrapolate variance as a learnable convex combination between the estimated heteroscedastic variance (inverted samples from the parameterized Gamma) and some pre-defined, larger, non-trainable variance. They perform ablation and find that their methods are complementary on three UCI regression tasks with the locality sampler and Student's $t$ distribution individually providing the most benefit. They report the combination of all their proposals generally outperforms their chosen baselines: Gaussian process regression (Williams & Rasmussen, 2006; Snelson & Ghahramani, 2006; Damianou & Lawrence, 2013), unmodified neural-network parameterizations of mean and variance (Nix & Weigend, 1994; Bishop, 1994; Kingma & Welling, 2013; Rezende et al., 2014), Bayesian neural networks (MacKay, 1992; Hernández-Lobato & Adams, 2015), and MC Drop Out (Gal & Ghahramani, 2016).

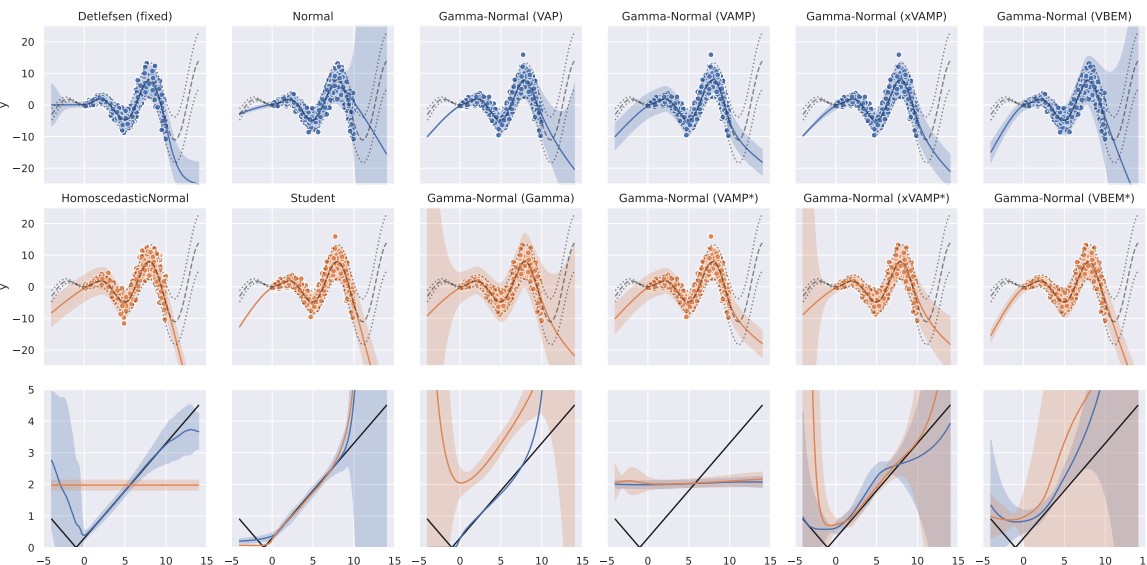

Figure 4: Toy regression results for the unmodified toy process (e.g. no isolated data point) from Detlefsen et al. (2019). The subplot descriptors are identical to fig. 1

### C.2 Precision's Exact Posterior

Employing amortized VI may seem superfluous, however, since the exact posterior, $p(\lambda|x, y)$, is available. However, the resulting predictive distribution's variance lacks dependence on $x_*$ rendering it homoscedastic. Thus, we forego posterior exactness for heteroscedasticity and the ability to probabilistically regularize variance. The following derivation shows that precision's true posterior for regression results in a distribution that depends both on the covariates $x_i$ and responses $y_i$. This dual dependence implies the true posterior falls outside the scope of heteroscedasticity due to the additional dependence on $y_i$ and limits predictive utility when $y_*$ is unobservable.

$$p(\lambda|y, x) = \frac{p(y, \lambda|x)}{p(y|x)} = \frac{p(y, \lambda|x)}{\int p(y, \lambda|x)d\lambda} = \frac{p(y|x, \lambda)p(\lambda)}{\int p(y, \lambda|x)d\lambda} = \frac{\prod_{i=1}^n \mathcal{N}(y_i|\mu(x_i), \lambda_i)p(\lambda_i)}{\int \prod_{i=1}^n \mathcal{N}(y_i|\mu(x_i), \lambda_i)p(\lambda_i)d\lambda_i}$$
$$= \frac{\prod_{i=1}^n \mathcal{N}(y_i|\mu(x_i), \lambda_i)p(\lambda_i)}{\prod_{i=1}^n \int \mathcal{N}(y_i|\mu(x_i), \lambda_i)p(\lambda_i)d\lambda_i} = \prod_{i=1}^n p(\lambda_i|y_i, x_i)$$

Above, we use $p(\lambda_i|y_i, x_i) \triangleq \frac{\mathcal{N}(y_i|\mu(x_i), \lambda_i)p(\lambda_i)}{\int \mathcal{N}(y_i|\mu(x_i), \lambda_i)p(\lambda_i)d\lambda_i}$ to symbolically capture the local factorization.

### C.3 Toy Data

In fig. 1, we set $\epsilon = 0$ at $x = 7.5$ to keep the rogue point in the same place across trials. Detlefsen et al. (2019) use single-layer neural networks with 50 sigmoid activations. We use the same size network but with ELU activations. We use ELU activations for all experiments, whereas Detlefsen et al. (2019) change activations for the UCI and VAE tasks. For the standard Gamma prior's parameters, we fit a MLE Gamma to the true precisions of the training data. For our VAMP(*), xVAMP(*), and VBEM* priors, we set $K = 20$. For VAMP(*) and xVAMP(*), we sample pseudo-inputs $u_i \overset{iid}{\sim} \text{Uniform}([-4, 14])$. Like Detlefsen et al. (2019), we use ADAM (Kingma & Ba, 2014) for optimization. While Detlefsen et al. (2019) employ separate optimizers for the mean and variance networks that respectively use 1e-2 and 1e-3 as learning rates, we employ a single ADAM instance with a learning rate of 5e-3. We run all algorithms for 6e3 epochs without batching (the single batch contains all 500 training points). We ran the toy experiments on a NVIDIA RTX2070.

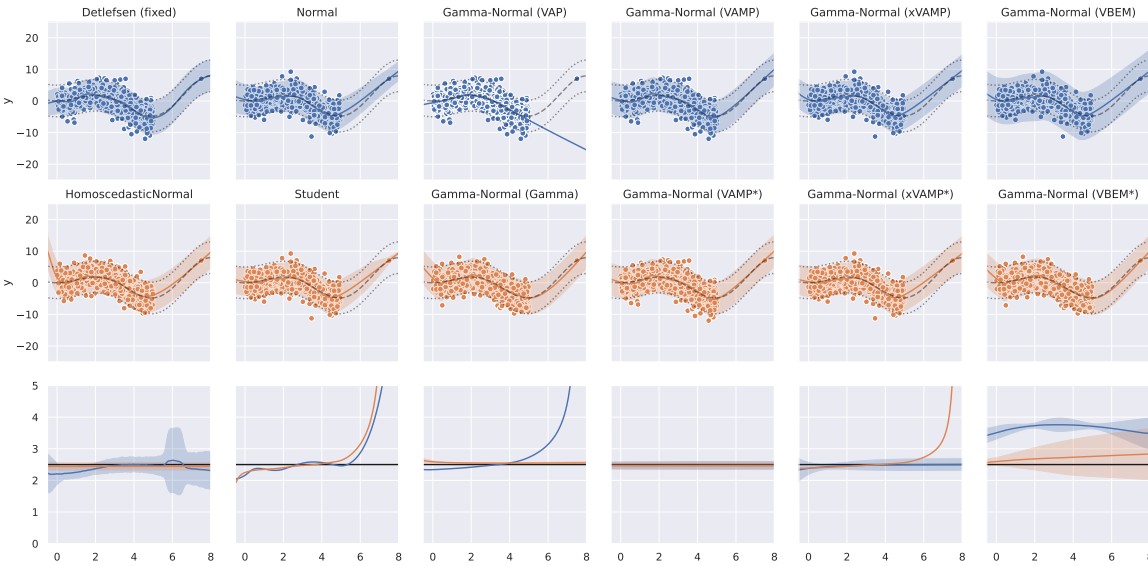

Figure 5: Toy regression results for a homoscedastic toy process with an isolated data point. The subplot descriptors are identical to fig. 1

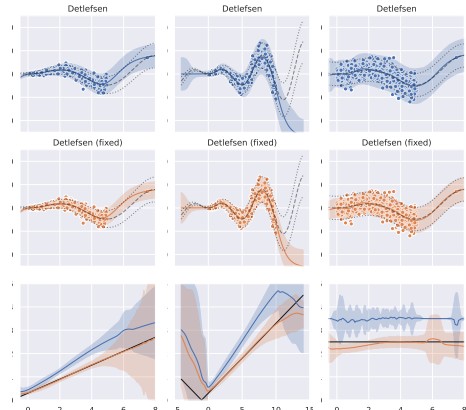

Figure 6: We compare the before and after effects of our bug fix for Detlefsen et al. (2019). From left to right, we respectively consider the generative processes from figs. 1, 4 and 5

In section 3.1, we modified the toy process from Detlefsen et al. (2019) to introduce an isolated data point in order to demonstrate the inability of the Normal, Student, and VAP baselines to converge on the target. There, these models employed high noise variance to explain the rogue point. Figure 4 contains results for the unmodified toy process where covariates are sampled from Uniform($[0, 10]$). Now, the under-performing baseline models do well since there are no rogue data points for which the mean network would experience exploding gradients.

Figure 5 contains results for our modified process (section 3.1), but now the true variance is homoscedastic. We find that the majority of our heteroscedastic models both converge on the isolated point at $x = 7.5$ and accurately model the true noise variance.

### C.4 UCI Data

For the UCI regression tasks, Detlefsen et al. (2019) employ single-layer neural networks with 50 ReLU activations. We use the same network architecture but with ELU activations, which we found to be more robust during our Monte-Carlo estimation of the right-most term of eq. (6)'s RHS. Detlefsen et al. (2019) allow training to run for some number of batch iterations, whereas our code uses the notion of an epoch, which encompasses the number of batches required to see each example in the training set exactly once. To keep things equal, we allow each algorithm to run for a dataset-specific number of batch iterations with a batch size of 256, which we convert to epochs ($\lceil \frac{\text{iterations}}{\text{batch size}} \rceil$) for our methods. All UCI datasets use 2e4 batch iterations except for those with larger ($N > 9000$) sample sizes (e.g. carbon, naval, power plant, and superconductivity), which use 1e5 batch iterations. Here, Detlefsen et al. (2019) use 1e-2 and 1e-4 as learning rates for the mean and variance networks' ADAM optimizers, respectively. We use 1e-3 as the learning rate for our single ADAM instance. We use $a = 1$ and $b = 0.001$ as the standard Gamma prior's parameters. For VAMP(*) and xVAMP(*) we sample $K = 100$ pseudo-inputs uniformly from the training set without replacement. We also use $K = 100$ for our VBEM* prior. We employ early stopping on the validation set's log (posterior) predictive likelihood with a patience of 50 epochs. We implemented an equivalent early stopping mechanism in the baseline code of Detlefsen et al. (2019), in which we also introduced support for multivariate response variables. We ran the UCI experiments on a NVIDIA RTX2070 and were able to parallelize up to five trials (i.e. five concurrent training sessions for any of the tested models)–our system ram (16GB) was the limiting factor.

Tables 5 to 8 each contain one of our four proposed PPCs. When available, we italicize any cited (i.e. reported) results. In these tables, we bold just the top performer, but never bold cited results since we did not validate these methods under our experimental conditions (e.g. some reported results use wider and/or deeper neural networks). Table 4 tallies the number of top PPC performances as well as statistical ties for top PPC performance across UCI datasets. We exclude cited results from table 4, but compare performances in the following paragraphs.

Sun et al. (2019) introduce functional variational Bayesian neural networks, which they test on some of the same UCI regression tasks that we consider. For power plant, they employ single hidden layer networks with 100 neurons, but otherwise use 50 neurons like we do. They report energy and naval metrics as well, but we suspect they only regressed one dimension of the two-dimensional targets. Unfortunately, their code loads *.data files for these two sets making it difficult for us to confirm our suspicion and identify the regressed dimension. As such, we exclude any multi-dimensional sets. Sun et al. (2019) provide mean, standard deviation, and the number of trials (10) for their metrics; this allows us to conduct a two-sided Welch's T-test against our Gamma-Normal VBEM* model. For log likelihood $p$-values, we obtain 0.088, 0.001, 0.012, and 0.539 respectively for boston, concrete, power plant, and yacht. Using a $p \leq 0.05$ threshold, VBEM* statistically beats and ties Sun et al. (2019) twice each. For mean RMSE $p$-values, we obtain 0.005, 0.033, 0.663, and 1.19 e-7 suggesting Sun et al. (2019) statistically outperforms VBEM*'s mean RMSE for three of these data sets.

Sicking et al. (2021) introduce a second-moment loss term to improve uncertainty estimation. They report negative log likelihood and mean RMSE for normalized target scalings (i.e. zero mean and unit variance). We multiplied their reported RMSEs by the target's standard deviation to attain RMSEs for the original target scalings, which we report. These rescaled RMSEs seemed reasonable compared to other results. However, adjusting their reported log likelihoods to the original target scalings produces nonsensible results that were too good to be true when compared to their RMSEs. Without access to their code, we cannot confirm that our adjustment by $\frac{1}{2} \log \sigma_{\text{target}}^2$ is appropriate. Furthermore, we exclude any results they report for multi-dimensional targets since rescaling likelihood is no longer straight forward. Recognizing Sicking et al. (2021) employ neural networks with two hidden layers, each with 50 ReLU activations, we implemented our VBEM* prior with these network sizes and denote it 'Gamma-Normal (2x)' in tables 5 to 8. Because this additional model uses larger networks and because we only tested it on the univariate UCI sets that overlap with Sicking et al. (2021), we never bold it in tables 5 to 8 and also exclude it from table 4. Examining table 6, we find that the single reported mean RMSE value from Sicking et al. (2021) is within two standard deviations of (similar to) Gamma-Normal (2x) VBEM* for four data sets, above two standard deviations (worse than) once (yacht), and below two standard deviations (better than) once (superconductivity).

Table 4: UCI regression summary: We tally the number of datasets for which a method was the top PPC performer or was statistically indistinguishable from the winner according to a two-sided Kolmogorov–Smirnov test with $p \leq 0.05$. Tallied statistical ties appear in parentheses.

| Algorithm | Prior | LL | Mean RMSE | Var Bias | Sample RMSE | Total |
|---|---|---|---|---|---|---|
| Detlefsen | N/A | 0 (0) | 1 (1) | 0 (0) | 0 (0) | 1 (1) |
| Normal | N/A | 0 (3) | 2 (7) | 1 (6) | **3** (6) | 6 (22) |
| Student | N/A | 3 (**7**) | 0 (6) | 0 (5) | 2 (5) | 5 (23) |
| Gamma-Normal | VAP | 0 (4) | 0 (6) | 0 (7) | 0 (5) | 0 (22) |
| | Gamma | 0 (0) | 0 (6) | **2** (6) | 0 (5) | 2 (17) |
| | VAMP | 0 (3) | 0 (8) | 1 (7) | 1 (**9**) | 2 (**27**) |
| | VAMP* | 0 (3) | 1 (8) | **2** (7) | 2 (**9**) | 5 (**27**) |
| | xVAMP | 1 (4) | 0 (7) | 1 (6) | 2 (6) | 4 (23) |
| | xVAMP* | 1 (4) | 0 (7) | 1 (**8**) | 0 (6) | 2 (25) |
| | VBEM | 0 (1) | **5** (**10**) | 0 (0) | 0 (0) | 5 (11) |
| | VBEM* | **5** (**7**) | 1 (7) | **2** (5) | 0 (5) | **8** (24) |

Table 5: UCI predictive log likelihood reported as mean±std. Tuples appearing below dataset are $(N_{\text{observations}}, \dim(x), \dim(y))$.

| Algorithm | Prior | boston (506, 13, 1) | carbon (10721, 5, 3) | concrete (1030, 8, 1) | energy (768, 8, 2) | naval (11934, 16, 2) |
|---|---|---|---|---|---|---|
| Sun et al. (2019) | N/A | *-2.30±0.04* | – | *-3.10±0.02* | – | – |
| Detlefsen | N/A | -2.98±0.09 | 8.77±0.24 | -3.66±0.08 | -4.90±0.27 | 9.67±0.19 |
| Normal | N/A | -2.42±0.23 | 13.20±1.35 | -3.06±0.17 | -0.48±0.69 | 14.15±0.17 |
| Student | N/A | -2.37±0.19 | **17.19±0.21** | -3.10±0.17 | 0.22±0.31 | 13.60±0.39 |
| Gamma-Normal | VAP | -2.36±0.17 | 15.52±0.24 | -3.12±0.17 | 0.17±0.44 | 13.36±0.41 |
| | Gamma | -2.48±0.29 | 11.28±0.02 | -3.20±0.16 | -1.05±0.18 | 12.33±0.16 |
| | VAMP | -2.39±0.17 | 14.37±0.17 | -3.09±0.16 | -0.18±0.21 | 14.16±0.78 |
| | VAMP* | -2.39±0.16 | 14.38±0.12 | -3.09±0.16 | -0.16±0.20 | 13.96±0.88 |
| | xVAMP | **-2.33±0.17** | 15.38±0.24 | -3.01±0.14 | 0.05±0.28 | 13.50±0.59 |
| | xVAMP* | -2.33±0.17 | 15.41±0.18 | -3.01±0.13 | 0.11±0.39 | 13.34±0.47 |
| | VBEM | -2.46±0.11 | 4.57±1.00 | -3.11±0.07 | -4.52±0.26 | 9.02±0.61 |
| | VBEM* | -2.36±0.14 | 14.64±0.16 | **-2.99±0.13** | **0.49±0.28** | **14.42±0.15** |
| Gamma-Normal (2x) | VBEM* | -2.31±0.17 | – | -2.88±0.14 | – | – |

| Algorithm | Prior | power plant (9568, 4, 1) | superconductivity (21263, 81, 1) | wine-red (1599, 11, 1) | wine-white (4898, 11, 1) | yacht (308, 6, 1) |
|---|---|---|---|---|---|---|
| Sun et al. (2019) | N/A | *-2.83±0.01* | – | – | – | *-1.03±0.03* |
| Detlefsen | N/A | -3.26±9.1e-03 | -5.21±0.02 | -1.04±0.06 | -1.12±0.04 | -3.15±0.10 |
| Normal | N/A | -2.82±0.05 | -3.51±0.10 | -0.92±0.05 | -1.05±0.04 | -1.55±0.65 |
| Student | N/A | **-2.78±0.03** | -3.41±0.05 | **-0.80±0.10** | -1.05±0.04 | -1.73±0.59 |
| Gamma-Normal | VAP | -2.81±0.04 | -3.45±0.06 | -0.87±0.06 | -1.04±0.04 | -1.79±0.50 |
| | Gamma | -2.88±0.03 | -3.45±0.04 | -0.98±0.07 | -1.13±0.05 | -1.73±0.38 |
| | VAMP | -2.83±0.03 | -3.94±0.02 | -0.94±0.05 | -1.05±0.04 | -2.83±0.70 |
| | VAMP* | -2.83±0.03 | -3.94±0.03 | -0.94±0.05 | -1.05±0.04 | -2.77±0.77 |
| | xVAMP | -2.81±0.04 | -3.40±0.04 | -0.90±0.05 | -1.03±0.04 | -1.68±0.38 |
| | xVAMP* | -2.81±0.04 | **-3.39±0.05** | -0.89±0.06 | -1.03±0.04 | -1.71±0.47 |
| | VBEM | -2.89±0.05 | -3.77±0.09 | -0.91±0.05 | -1.03±0.03 | -2.64±0.23 |
| | VBEM* | -2.81±0.03 | -3.41±0.04 | -0.89±0.06 | **-1.03±0.04** | **-1.11±0.57** |
| Gamma-Normal (2x) | VBEM* | -2.76±0.03 | -3.31±0.03 | -0.89±0.06 | – | -0.91±0.29 |

Table 6: UCI predictive mean RMSE reported as mean±std. Tuples appearing below dataset are $(N_{\text{observations}}, \dim(x), \dim(y))$.

| Algorithm | Prior | boston (506, 13, 1) | carbon (10721, 5, 3) | concrete (1030, 8, 1) | energy (768, 8, 2) | naval (11934, 16, 2) |
|---|---|---|---|---|---|---|
| Sun et al. (2019) | N/A | *2.38±0.10* | – | *4.94±0.18* | – | – |
| Sicking et al. (2021) | N/A | *3.03* | – | *4.17* | – | – |
| Detlefsen | N/A | 4.48±1.06 | 0.02±4.6e-03 | 8.13±1.65 | 2.05±0.49 | 4.2e-03±6.3e-04 |
| Normal | N/A | 3.36±1.29 | **7.5e-03±3.3e-03** | 6.05±0.66 | 1.30±0.14 | 3.5e-03±3.1e-04 |
| Student | N/A | 3.62±1.42 | 7.6e-03±3.3e-03 | 6.71±0.81 | 1.42±0.17 | 3.4e-03±5.0e-04 |
| Gamma-Normal | VAP | 3.44±1.21 | 7.7e-03±3.3e-03 | 6.61±0.84 | 1.38±0.15 | 3.2e-03±5.3e-04 |
| | Gamma | 3.82±1.72 | 7.6e-03±3.3e-03 | 6.63±0.70 | 1.31±0.14 | 3.2e-03±5.1e-04 |
| | VAMP | 3.15±1.06 | 7.8e-03±3.2e-03 | 5.47±1.00 | 1.36±0.13 | 1.2e-03±1.0e-03 |
| | VAMP* | 3.15±1.05 | 7.8e-03±3.2e-03 | 5.47±1.00 | 1.36±0.13 | 1.6e-03±1.3e-03 |
| | xVAMP | 3.25±1.16 | 7.6e-03±3.3e-03 | 5.61±0.67 | 1.36±0.14 | 3.2e-03±5.2e-04 |
| | xVAMP* | 3.28±1.17 | 7.6e-03±3.3e-03 | 5.72±0.59 | 1.36±0.14 | 3.2e-03±4.9e-04 |
| | VBEM | **3.14±1.07** | 8.7e-03±3.3e-03 | **5.26±0.58** | 1.36±0.14 | **5.6e-04±1.6e-04** |
| | VBEM* | 3.18±1.12 | 7.6e-03±3.3e-03 | 5.59±0.70 | **1.30±0.13** | 2.4e-03±2.8e-04 |
| Gamma-Normal (2x) | VBEM* | 3.04±1.14 | – | 5.00±0.59 | – | – |

| Algorithm | Prior | power plant (9568, 4, 1) | superconductivity (21263, 81, 1) | wine-red (1599, 11, 1) | wine-white (4898, 11, 1) | yacht (308, 6, 1) |
|---|---|---|---|---|---|---|
| Sun et al. (2019) | N/A | *4.10±0.05* | – | – | – | *0.61±0.07* |
| Sicking et al. (2021) | N/A | *3.75* | *10.96* | *0.65* | – | *1.21* |
| Detlefsen | N/A | 4.33±0.27 | 17.72±1.29 | 0.71±0.06 | 0.76±0.04 | **2.42±1.06** |
| Normal | N/A | **4.12±0.20** | 14.53±0.44 | 0.62±0.03 | 0.70±0.04 | 3.42±2.30 |
| Student | N/A | 4.12±0.19 | 14.85±0.42 | 0.63±0.03 | 0.71±0.03 | 15.03±3.30 |
| Gamma-Normal | VAP | 4.14±0.21 | 14.83±0.48 | 0.62±0.03 | 0.70±0.03 | 14.70±3.31 |
| | Gamma | 4.18±0.18 | 14.44±0.43 | 0.63±0.03 | 0.72±0.03 | 12.17±2.38 |
| | VAMP | 4.16±0.20 | 12.81±0.33 | 0.62±0.03 | 0.70±0.04 | 5.42±3.54 |
| | VAMP* | 4.16±0.20 | **12.80±0.35** | 0.62±0.03 | 0.70±0.04 | 5.30±3.65 |
| | xVAMP | 4.14±0.20 | 14.13±0.39 | 0.62±0.03 | 0.70±0.04 | 12.30±3.09 |
| | xVAMP* | 4.13±0.21 | 14.25±0.42 | 0.62±0.03 | 0.70±0.03 | 12.51±3.20 |
| | VBEM | 4.16±0.19 | 13.13±0.37 | **0.62±0.03** | **0.69±0.03** | 3.51±1.46 |
| | VBEM* | 4.12±0.19 | 14.08±0.42 | 0.62±0.03 | 0.69±0.03 | 5.33±2.58 |
| Gamma-Normal (2x) | VBEM* | 3.92±0.20 | 12.71±0.34 | 0.62±0.03 | – | 0.47±0.18 |

Table 7: UCI predictive variance bias reported as mean±std. Tuples appearing below dataset are $(N_{\text{observations}}, \dim(x), \dim(y))$.

| Algorithm | Prior | boston (506, 13, 1) | carbon (10721, 5, 3) | concrete (1030, 8, 1) | energy (768, 8, 2) | naval (11934, 16, 2) |
|---|---|---|---|---|---|---|
| Detlefsen | N/A | 1.0e+02±79.11 | 9.8e-05±1.6e-04 | 2.2e+02±91.85 | 18.60±8.88 | nan±nan |
| Normal | N/A | 31.63±1.5e+02 | 3.5e+23±1.6e+24 | -2.01±8.67 | -0.16±0.24 | 3.1e-07±2.0e-06 |
| Student | N/A | 18.08±63.79 | 0.12±0.23 | -2.20±9.28 | 24.00±85.42 | 4.9e-06±2.2e-05 |
| Gamma-Normal | VAP | 3.3e+02±1.3e+03 | 0.25±1.11 | -2.13±7.85 | 0.04±0.31 | 3.1e-07±6.8e-07 |
| | Gamma | 3.18±20.09 | 1.5e-04±5.8e-05 | 0.76±11.35 | 0.31±0.40 | 3.7e-06±2.9e-06 |
| | VAMP | -2.96±7.96 | -6.6e-06±6.0e-05 | -6.15±5.18 | -0.15±0.39 | 1.7e-07±2.9e-07 |
| | VAMP* | -3.00±7.84 | **-6.0e-06±6.0e-05** | -6.16±5.18 | -0.13±0.40 | **1.3e-07±3.3e-07** |
| | xVAMP | 0.65±16.20 | 2.7e-05±9.7e-05 | -4.82±4.63 | **6.0e-03±0.36** | 3.1e-07±7.4e-07 |
| | xVAMP* | 0.51±20.17 | 5.0e-04±2.2e-03 | -4.66±5.07 | -8.5e-03±0.36 | 2.7e-07±6.3e-07 |
| | VBEM | 6.74±8.48 | 0.01±4.5e-03 | 25.86±8.94 | 22.06±5.58 | 3.6e-05±1.4e-05 |
| | VBEM* | **-0.11±8.62** | -7.2e-06±6.1e-05 | **-0.58±5.05** | 0.02±0.28 | 3.9e-07±4.9e-07 |
| Gamma-Normal (2x) | VBEM* | -1.60±6.98 | – | -2.69±4.29 | – | – |

| Algorithm | Prior | power plant (9568, 4, 1) | superconductivity (21263, 81, 1) | wine-red (1599, 11, 1) | wine-white (4898, 11, 1) | yacht (308, 6, 1) |
|---|---|---|---|---|---|---|
| Detlefsen | N/A | 69.25±2.40 | 5.5e+04±6.2e+03 | 2.16±1.57 | 0.83±0.36 | 96.62±54.08 |
| Normal | N/A | **0.05±1.53** | 2.3e+13±1.0e+14 | -3.8e-03±0.04 | -0.02±0.06 | 20.68±54.95 |
| Student | N/A | -0.27±1.47 | 1.6e+05±3.3e+05 | 12.52±30.71 | -5.6e-03±0.05 | 1.7e+03±2.3e+03 |
| Gamma-Normal | VAP | 0.52±1.29 | 9.0e+05±2.6e+06 | 0.03±0.05 | 0.13±0.64 | 1.3e+03±1.5e+03 |
| | Gamma | 2.34±1.43 | 1.1e+02±81.21 | 0.04±0.11 | **-2.3e-03±0.05** | **-7.28±40.88** |
| | VAMP | 0.89±1.04 | **-9.83±7.97** | 0.05±0.06 | -8.8e-03±0.04 | 38.05±83.39 |
| | VAMP* | 0.89±1.04 | -9.89±7.98 | 0.05±0.06 | -8.9e-03±0.04 | 38.07±83.29 |
| | xVAMP | 0.46±1.25 | 14.40±42.90 | 3.5e-03±0.05 | -0.03±0.03 | 4.8e+02±1.7e+03 |
| | xVAMP* | 0.44±1.24 | 1.3e+02±4.7e+02 | **2.1e-03±0.05** | -0.03±0.03 | 1.7e+02±1.5e+02 |
| | VBEM | 16.53±9.32 | 91.44±25.39 | 0.08±0.04 | 0.07±0.04 | 20.70±25.23 |
| | VBEM* | 1.86±1.44 | 9.87±16.22 | 0.05±0.06 | 0.01±0.04 | 26.48±26.88 |
| Gamma-Normal (2x) | VBEM* | 1.33±1.34 | 59.40±1.3e+02 | 0.04±0.05 | – | 0.86±0.55 |

Table 8: UCI predictive sample RMSE reported as mean±std. Tuples appearing below dataset are $(N_{\text{observations}}, \dim(x), \dim(y))$.

| Algorithm | Prior | boston (506, 13, 1) | carbon (10721, 5, 3) | concrete (1030, 8, 1) | energy (768, 8, 2) | naval (11934, 16, 2) |
|---|---|---|---|---|---|---|
| Detlefsen | N/A | 12.02±3.89 | 0.03±3.6e-03 | 17.93±2.55 | 5.07±0.98 | 6.2e-03±5.7e-04 |
| Normal | N/A | 4.92±3.57 | 2.6e+11±1.2e+12 | 8.23±1.08 | **1.85±0.21** | 5.0e-03±5.4e-04 |
| Student | N/A | 4.64±1.10 | **8.1e-03±3.1e-03** | 9.18±1.36 | 2.07±0.37 | 5.0e-03±1.7e-03 |
| Gamma-Normal | VAP | 4.69±0.86 | 0.01±2.2e-03 | 9.42±1.82 | 2.02±0.35 | 4.5e-03±7.2e-04 |
| | Gamma | 4.92±2.18 | 0.02±1.8e-03 | 8.67±1.20 | 1.88±0.38 | 4.6e-03±9.1e-04 |
| | VAMP | 4.27±0.87 | 0.01±1.9e-03 | 7.27±1.05 | 1.93±0.19 | **1.8e-03±1.4e-03** |
| | VAMP* | 4.26±0.86 | 0.01±2.0e-03 | **7.27±1.05** | 1.93±0.20 | 2.2e-03±1.8e-03 |
| | xVAMP | **4.23±1.15** | 0.01±2.0e-03 | 7.84±1.05 | 1.88±0.29 | 4.5e-03±7.5e-04 |
| | xVAMP* | 4.23±1.14 | 0.01±2.5e-03 | 8.00±1.01 | 1.87±0.30 | 4.5e-03±7.1e-04 |
| | VBEM | 5.03±0.92 | 0.11±0.03 | 9.21±0.98 | 5.13±0.83 | 5.9e-03±1.5e-03 |
| | VBEM* | 4.41±1.07 | 0.01±1.9e-03 | 7.90±1.10 | 1.85±0.30 | 3.5e-03±3.7e-04 |
| Gamma-Normal (2x) | VBEM* | 4.03±0.88 | – | 6.96±0.83 | – | – |

| Algorithm | Prior | power plant (9568, 4, 1) | superconductivity (21263, 81, 1) | wine-red (1599, 11, 1) | wine-white (4898, 11, 1) | yacht (308, 6, 1) |
|---|---|---|---|---|---|---|
| Detlefsen | N/A | 10.36±0.28 | 2.4e+02±15.98 | 1.43±0.34 | 1.27±0.10 | 10.71±2.19 |
| Normal | N/A | 5.85±0.20 | 1.7e+06±7.4e+06 | **0.86±0.07** | 0.98±0.03 | **4.73±3.68** |
| Student | N/A | **5.79±0.28** | 21.25±1.46 | 0.88±0.06 | 0.99±0.04 | 20.24±7.84 |
| Gamma-Normal | VAP | 5.90±0.26 | 21.05±0.81 | 0.89±0.06 | 0.99±0.04 | 20.00±7.47 |
| | Gamma | 6.01±0.51 | 23.75±13.27 | 0.90±0.13 | 0.98±0.06 | 14.20±4.40 |
| | VAMP | 5.97±0.28 | 17.86±0.41 | 0.90±0.07 | 0.99±0.06 | 8.67±6.43 |
| | VAMP* | 5.97±0.28 | **17.85±0.42** | 0.90±0.07 | 0.99±0.06 | 8.50±6.57 |
| | xVAMP | 5.92±0.20 | 19.98±0.49 | 0.89±0.05 | **0.97±0.06** | 15.57±5.76 |
| | xVAMP* | 5.91±0.20 | 20.19±0.77 | 0.89±0.05 | 0.97±0.06 | 15.81±5.05 |
| | VBEM | 7.17±0.65 | 20.92±0.70 | 0.93±0.06 | 1.02±0.05 | 6.66±2.53 |
| | VBEM* | 6.00±0.20 | 19.78±0.48 | 0.89±0.07 | 0.97±0.03 | 6.84±4.58 |
| Gamma-Normal (2x) | VBEM* | 5.67±0.23 | 18.09±0.49 | 0.89±0.07 | – | 1.08±0.31 |

Comparing Gamma-Normal (2x) VBEM* PPC performance in tables 5 to 8 to Gamma-Normal VBEM*, which uses shallower neural networks, is evidence our methods improve as parameter map flexibility increases. This observation suggests variance is still well regularized away from zero even as model flexibility increases.

**Active Learning**  We consider the same active learning regime from Detlefsen et al. (2019). We split each data set into 20% train, 60% reserve, and 20% test. The first active learning step utilizes the 20% training split. Thereafter, we move the $n$ points from the reserve pool with highest predicted variance to the training set. We define $n$ to be 5% of the original size of the reserve pool. We repeat this process ten times for each experiment and repeat each experiment ten times per data set. We preserve the remaining implementation details from the fully-supervised regression experiments, with two exceptions. First, we grow $K$, the number of mixture components available to the (x)VAMP(*) and VBEM(*) priors, proportionally to the ratio of utilized training data to total available. Specifically, we multiply $K = 100$ by this ratio at each active learning step to set the number of mixture components. Second, we identically scale the maximum allowed mini-batch iterations at each active learning step.

We plot the log predictive likelihoods and predictive mean RMSE on the held out test set across active learning steps in figs. 7 and 8. For clarity, we integrate these curves to reduce performance to a scalar for each dataset-method pair (tables 9 and 10). We find that VBEM* generally reigns supreme, which makes sense given its previous top performances on these data sets. Interestingly, we find cases for all methods, but on differing data sets, where additional training data does not improve test-set performance.

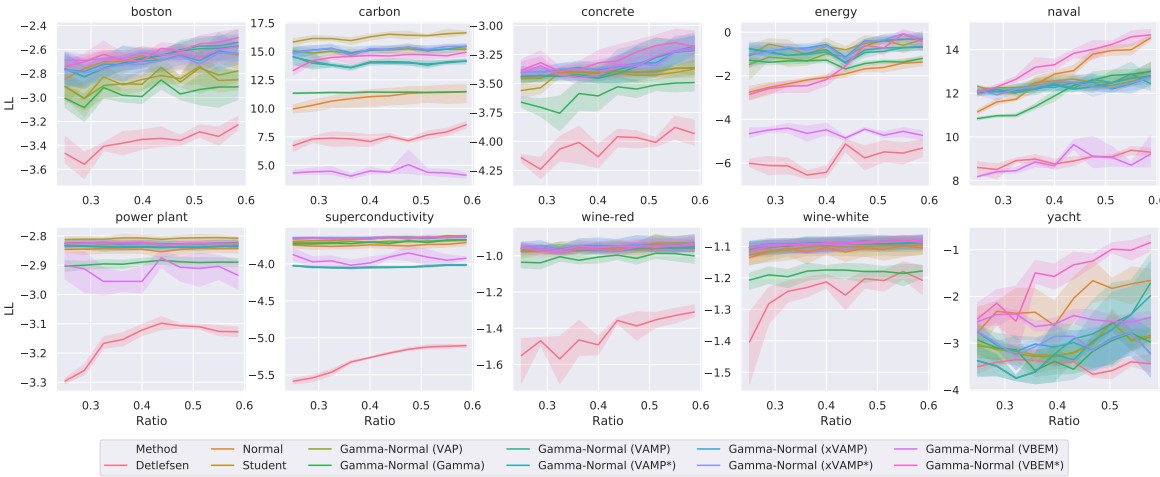

Figure 7: Log predictive likelihood across active learning steps for UCI data sets. The $x$ axis is the ratio of utilized training data to the available. Darker lines are mean performance across trials. Shaded areas denote 95% confidence intervals.

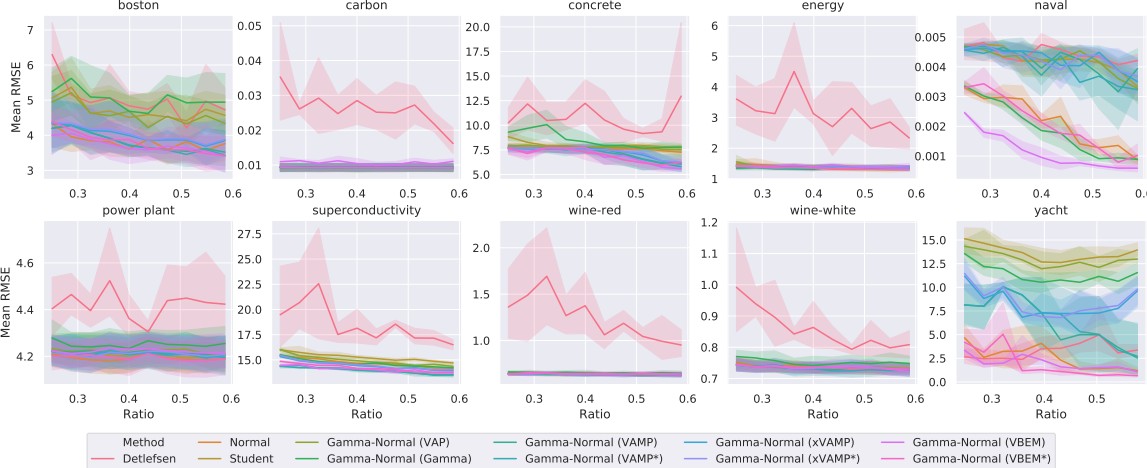

Figure 8: RMSE (of the predictive mean) across active learning steps for UCI data sets. The $x$ axis is the ratio of utilized training data to the available. Darker lines are mean performance across trials. Shaded areas denote 95% confidence intervals.

Table 9: UCI cumulative sum of log predictive likelihood across active learning steps reported as mean±std. We bold only the top performer. Tuples appearing below dataset are $(N_{\text{observations}}, \dim(x), \dim(y))$.

| Algorithm | Prior | boston (506, 13, 1) | carbon (10721, 5, 3) | concrete (1030, 8, 1) | energy (768, 8, 2) | naval (11934, 16, 2) |
|---|---|---|---|---|---|---|
| Detlefsen | N/A | -33.70±0.56 | 74.68±1.54 | -40.33±0.27 | -58.53±1.33 | 89.54±0.64 |
| Normal | N/A | -26.92±1.30 | 1.1e+02±8.71 | -34.03±0.75 | -19.93±1.82 | 1.3e+02±1.03 |
| Student | N/A | -28.60±0.77 | **1.6e+02±0.91** | -34.41±0.47 | -6.67±2.24 | 1.2e+02±0.98 |
| Gamma-Normal | VAP | -28.56±0.95 | 1.5e+02±2.78 | -34.06±0.31 | -8.90±2.16 | 1.2e+02±0.73 |
| | Gamma | -29.57±0.75 | 1.1e+02±0.32 | -35.90±0.81 | -13.62±1.34 | 1.2e+02±1.36 |
| | VAMP | -26.76±0.70 | 1.4e+02±2.50 | -33.69±1.06 | -9.10±1.35 | 1.2e+02±1.70 |
| | VAMP* | -26.66±0.76 | 1.4e+02±2.46 | -33.57±1.08 | -9.02±1.10 | 1.2e+02±0.94 |
| | xVAMP | -26.98±1.20 | 1.5e+02±4.01 | -33.36±0.89 | -6.59±1.70 | 1.2e+02±0.70 |
| | xVAMP* | -27.10±1.19 | 1.5e+02±3.29 | -33.39±0.89 | **-6.43±1.29** | 1.2e+02±0.76 |
| | VBEM | -26.48±0.53 | 44.05±2.86 | -33.43±0.54 | -46.01±0.83 | 88.31±2.09 |
| | VBEM* | **-26.27±0.66** | 1.4e+02±3.85 | **-32.84±1.09** | -15.95±1.04 | **1.3e+02±1.53** |

| Algorithm | Prior | power plant (9568, 4, 1) | superconductivity (21263, 81, 1) | wine-red (1599, 11, 1) | wine-white (4898, 11, 1) | yacht (308, 6, 1) |
|---|---|---|---|---|---|---|
| Detlefsen | N/A | -31.57±0.06 | -52.89±0.20 | -14.28±0.48 | -12.43±0.28 | -34.55±1.28 |
| Normal | N/A | -28.45±0.19 | -37.46±0.52 | -9.66±0.30 | -11.12±0.30 | -21.33±2.80 |
| Student | N/A | **-28.08±0.12** | -36.57±0.07 | -9.49±0.31 | -11.05±0.33 | -30.38±1.15 |
| Gamma-Normal | VAP | -28.25±0.13 | -36.99±0.07 | -9.56±0.39 | -10.95±0.29 | -30.52±0.91 |
| | Gamma | -28.93±0.19 | -37.08±0.10 | -10.10±0.41 | -11.85±0.33 | -31.59±1.64 |
| | VAMP | -28.37±0.12 | -40.36±0.10 | -9.61±0.30 | -10.95±0.27 | -29.86±4.92 |
| | VAMP* | -28.36±0.12 | -40.37±0.06 | -9.64±0.29 | -10.92±0.27 | -30.87±4.02 |
| | xVAMP | -28.26±0.13 | -36.48±0.05 | **-9.46±0.43** | **-10.89±0.27** | -30.11±2.68 |
| | xVAMP* | -28.26±0.13 | **-36.42±0.04** | -9.48±0.42 | -10.89±0.28 | -29.85±1.34 |
| | VBEM | -29.21±0.10 | -39.35±0.37 | -9.53±0.35 | -10.92±0.25 | -24.99±0.77 |
| | VBEM* | -28.26±0.14 | -36.50±0.07 | -9.59±0.36 | -10.93±0.26 | **-15.64±1.91** |

Table 10: UCI cumulative sum of RMSE (of predictive mean) across active learning steps reported as mean±std. We bold only the top performer. Tuples appearing below dataset are $(N_{\text{observations}}, \dim(x), \dim(y))$.

| Algorithm | Prior | boston (506, 13, 1) | carbon (10721, 5, 3) | concrete (1030, 8, 1) | energy (768, 8, 2) | naval (11934, 16, 2) |
|---|---|---|---|---|---|---|
| Detlefsen | N/A | 49.96±3.89 | 0.26±0.04 | 1.1e+02±13.71 | 31.41±4.79 | 0.04±1.4e-03 |
| Normal | N/A | 38.17±6.22 | 0.09±0.01 | 75.95±5.43 | 13.68±0.99 | 0.02±1.2e-03 |
| Student | N/A | 47.08±7.00 | **0.09±0.01** | 80.01±2.80 | 14.14±0.85 | 0.04±1.3e-03 |
| Gamma-Normal | VAP | 46.02±6.50 | 0.09±0.01 | 77.75±3.04 | 13.98±0.78 | 0.04±1.7e-03 |
| | Gamma | 50.29±6.97 | 0.09±0.01 | 85.00±5.72 | **13.41±0.64** | 0.02±7.9e-04 |
| | VAMP | 38.11±5.99 | 0.09±0.01 | 71.79±7.33 | 13.95±0.63 | 0.04±3.3e-03 |
| | VAMP* | 37.88±6.21 | 0.09±0.01 | 70.86±7.23 | 13.99±0.62 | 0.04±2.5e-03 |
| | xVAMP | 39.99±7.30 | 0.09±0.01 | 72.43±5.52 | 13.74±0.82 | 0.04±1.2e-03 |
| | xVAMP* | 40.36±7.08 | 0.09±0.01 | 72.82±5.74 | 13.67±0.78 | 0.04±1.6e-03 |
| | VBEM | **37.17±5.66** | 0.11±0.01 | 69.39±5.31 | 14.20±0.63 | **0.01±1.1e-03** |
| | VBEM* | 37.93±5.86 | 0.09±0.01 | **69.25±6.34** | 13.50±0.63 | 0.02±1.3e-03 |

| Algorithm | Prior | power plant (9568, 4, 1) | superconductivity (21263, 81, 1) | wine-red (1599, 11, 1) | wine-white (4898, 11, 1) | yacht (308, 6, 1) |
|---|---|---|---|---|---|---|
| Detlefsen | N/A | 44.20±0.76 | 1.8e+02±8.63 | 12.42±1.50 | 8.58±0.39 | 32.56±6.07 |
| Normal | N/A | **41.93±0.81** | 1.5e+02±1.62 | 6.36±0.19 | 7.37±0.22 | 25.62±9.98 |
| Student | N/A | 42.09±0.86 | 1.5e+02±1.73 | 6.41±0.17 | 7.37±0.20 | 1.4e+02±11.74 |
| Gamma-Normal | VAP | 42.21±0.85 | 1.5e+02±0.96 | 6.38±0.20 | 7.35±0.20 | 1.3e+02±10.75 |
| | Gamma | 42.51±0.76 | 1.5e+02±0.69 | 6.52±0.22 | 7.53±0.30 | 1.1e+02±5.96 |
| | VAMP | 42.14±0.92 | 1.4e+02±1.32 | 6.39±0.23 | 7.35±0.24 | 63.26±22.50 |
| | VAMP* | 42.11±0.93 | **1.4e+02±0.72** | 6.40±0.22 | **7.32±0.23** | 66.27±19.49 |
| | xVAMP | 42.09±0.92 | 1.4e+02±0.59 | **6.36±0.22** | 7.35±0.22 | 82.86±9.68 |
| | xVAMP* | 42.11±0.89 | 1.5e+02±0.52 | 6.37±0.22 | 7.36±0.22 | 84.89±7.17 |
| | VBEM | 42.22±0.79 | 1.4e+02±0.58 | 6.39±0.22 | 7.35±0.23 | 19.66±2.97 |
| | VBEM* | 41.94±0.88 | 1.4e+02±0.91 | 6.44±0.24 | 7.36±0.21 | **18.97±4.82** |

# D   Variational Variance for VAEs

Figure 9 depicts the graphical models for the various VAE methods. For V3VAE, one can draw a solid arrow from $z_i$ to $\lambda_i$ without introducing a generative cycle, thus confirming the validity of $p(\lambda_i|z_i)$ as a

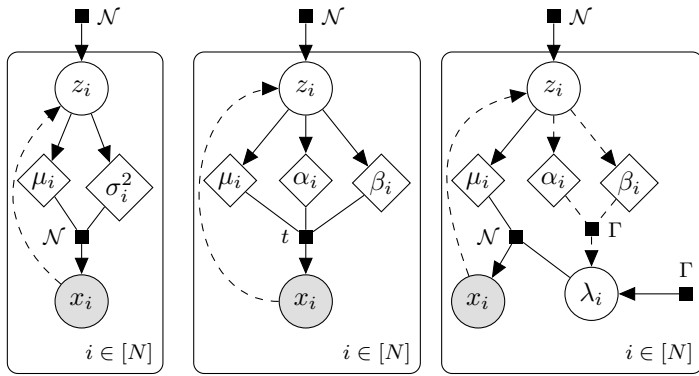

Figure 9: Graphical models for VAEs: Normal, Student's $t$, and Variational Variance VAE (left to right). Diamonds are deterministic neural network parameter maps. Solid arrows denote the generative process. Dashed arrows define the variational family.

heteroscedastic prior. For these experiments, we use ADAM with a 5e-5 learning rate. All Monte-Carlo (MC) approximations use 20 samples. We found additional samples did not improve log local reconstruction likelihood approximations. Since our VAMP(*), xVAMP(*), and VBEM* priors require twice as many MC samples ($q(\lambda|x)$ in addition to $q(z|x)$), their memory footprint is higher, requiring a batch size of 125 on a NVIDIA RTX2070. The remaining models use a batch size of 256. Because the lower batch size has twice as many batch updates per epoch, those models train for half (500) the number of epochs used by the other models (1000). For Celeb a, we further reduce the number of training epochs by half given the larger number of training examples (i.e. more batch iterations per epoch). We employ early stopping on the validation set's log posterior predictive probability with a patience of 25 for the 500 epoch models and 50 for the 1000 epoch models. We use an encoder architecture with hidden layers of sizes 512, 256, and 128, each of which applies an ELU activation. The decoder architecture is the transpose of the encoder. Because we only consider dense architectures, we had to resize Celeb-a images to $32 \times 26$ pixels in order to fit the weight matrices in our NVIDIA RTX2070's available VRAM. The dimensions of the latent variable, $\dim(z)$, are 10 for MNIST, 25 for Fashion MNIST, and 50 for Celeb-a. We include PPC metrics for MNIST and Celeb-a in tables 11 and 12, which we could not fit in the main report. Additionally, we include figs. 10 to 12, which are similar to fig. 9 (main article) but have additional samples for *all* tested methods.

Table 11: VAE PPCs for MNIST (mean±std.)

| Method | LL | Mean RMSE | Var Bias | Sample RMSE |
|---|---|---|---|---|
| Fixed-Var. VAE (1.0) | -732.10±0.11 | 0.17±1.1e-03 | 0.98±4.2e-04 | 1.02±2.5e-04 |
| Fixed-Var. VAE (0.001) | -2902.66±29.23 | **0.11±3.3e-04** | -1.2e-02±7.4e-05 | **0.12±3.1e-04** |
| VAE | 2593.51±267.72 | 0.25±2.7e-03 | 4.3e-02±2.6e-02 | 0.41±3.5e-02 |
| VAE + BN | 2386.70±23.17 | 0.25±1.8e-03 | 0.13±2.6e-02 | 0.50±2.6e-02 |
| VAE-Split | 2282.32±65.63 | 0.25±2.7e-03 | 7.4e-02±2.6e-02 | 0.44±3.2e-02 |
| VAE-Split + BN | 2482.36±75.34 | 0.28±4.4e-03 | 6.2e-02±1.1e-02 | 0.47±1.5e-02 |
| Detlefsen (0.001) | -1063.51±83.32 | 0.17±3.9e-04 | 15.03±3.39 | 3.86±0.51 |
| Detlefsen (0.25) | -168.23±2.39 | 0.17±4.6e-04 | 0.20±1.2e-03 | 0.51±9.6e-04 |
| Detlefsen (10.0) | -1556.55±1.72 | 0.17±5.4e-04 | 9.09±2.2e-02 | 3.02±4.0e-03 |
| MAP-VAE | 1291.42±6.94 | 0.13±1.9e-03 | -1.3e-02±4.1e-04 | 0.15±2.0e-03 |
| Student-VAE | **4826.82±530.95** | 0.27±1.7e-02 | 0.38±0.45 | 0.68±0.28 |
| V3AE-VAP | 3243.11±445.47 | 0.24±3.5e-03 | **8.1e-04±9.7e-04** | 0.34±5.5e-03 |
| V3AE-Gamma | 1495.01±2.75 | 0.13±7.0e-04 | -1.2e-02±1.9e-04 | 0.15±9.2e-04 |
| V3AE-VAMP | 2355.12±13.40 | 0.20±6.7e-04 | **6.2e-04±1.1e-03** | 0.28±1.7e-03 |
| V3AE-VAMP* | 2270.76±41.89 | 0.20±7.9e-04 | **1.2e-03±1.1e-03** | 0.29±2.2e-03 |
| V3AE-xVAMP | 2323.38±94.35 | 0.20±2.6e-03 | **1.9e-03±6.8e-04** | 0.29±3.2e-03 |
| V3AE-xVAMP* | 2280.13±48.29 | 0.20±2.0e-03 | **6.5e-04±7.2e-04** | 0.29±3.7e-03 |
| V3AE-VBEM | 296.95±0.92 | 0.12±8.1e-04 | 6.1e-02±2.7e-04 | 0.30±2.7e-04 |
| V3AE-VBEM* | 2107.63±5.44 | 0.14±1.2e-03 | **1.6e-03±1.1e-04** | 0.20±1.6e-03 |

Table 12: VAE PPCs for Celeb-a (mean±std.)

| Method | LL | Mean RMSE | Var Bias | Sample RMSE |
|---|---|---|---|---|
| Fixed-Var. VAE (1.0) | -732.10±0.11 | 0.17±1.1e-03 | 0.98±4.2e-04 | 1.02±2.5e-04 |
| Fixed-Var. VAE (0.001) | -2902.66±29.23 | **0.11±3.3e-04** | -1.2e-02±7.4e-05 | **0.12±3.1e-04** |
| VAE | 2593.51±267.72 | 0.25±2.7e-03 | 4.3e-02±2.6e-02 | 0.41±3.5e-02 |
| VAE + BN | 2386.70±23.17 | 0.25±1.8e-03 | 0.13±2.6e-02 | 0.50±2.6e-02 |
| VAE-Split | 2282.32±65.63 | 0.25±2.7e-03 | 7.4e-02±2.6e-02 | 0.44±3.2e-02 |
| VAE-Split + BN | 2482.36±75.34 | 0.28±4.4e-03 | 6.2e-02±1.1e-02 | 0.47±1.5e-02 |
| Detlefsen (0.001) | -1063.51±83.32 | 0.17±3.9e-04 | 15.03±3.39 | 3.86±0.51 |
| Detlefsen (0.25) | -168.23±2.39 | 0.17±4.6e-04 | 0.20±1.2e-03 | 0.51±9.6e-04 |
| Detlefsen (10.0) | -1556.55±1.72 | 0.17±5.4e-04 | 9.09±2.2e-02 | 3.02±4.0e-03 |
| MAP-VAE | 1291.42±6.94 | 0.13±1.9e-03 | -1.3e-02±4.1e-04 | 0.15±2.0e-03 |
| Student-VAE | **4826.82±530.95** | 0.27±1.7e-02 | 0.38±0.45 | 0.68±0.28 |
| V3AE-VAP | 3243.11±445.47 | 0.24±3.5e-03 | **8.1e-04±9.7e-04** | 0.34±5.5e-03 |
| V3AE-Gamma | 1495.01±2.75 | 0.13±7.0e-04 | -1.2e-02±1.9e-04 | 0.15±9.2e-04 |
| V3AE-VAMP | 2355.12±13.40 | 0.20±6.7e-04 | **6.2e-04±1.1e-03** | 0.28±1.7e-03 |
| V3AE-VAMP* | 2270.76±41.89 | 0.20±7.9e-04 | **1.2e-03±1.1e-03** | 0.29±2.2e-03 |
| V3AE-xVAMP | 2323.38±94.35 | 0.20±2.6e-03 | **1.9e-03±6.8e-04** | 0.29±3.2e-03 |
| V3AE-xVAMP* | 2280.13±48.29 | 0.20±2.0e-03 | **6.5e-04±7.2e-04** | 0.29±3.7e-03 |
| V3AE-VBEM | 296.95±0.92 | 0.12±8.1e-04 | 6.1e-02±2.7e-04 | 0.30±2.7e-04 |
| V3AE-VBEM* | 2107.63±5.44 | 0.14±1.2e-03 | **1.6e-03±1.1e-04** | 0.20±1.6e-03 |

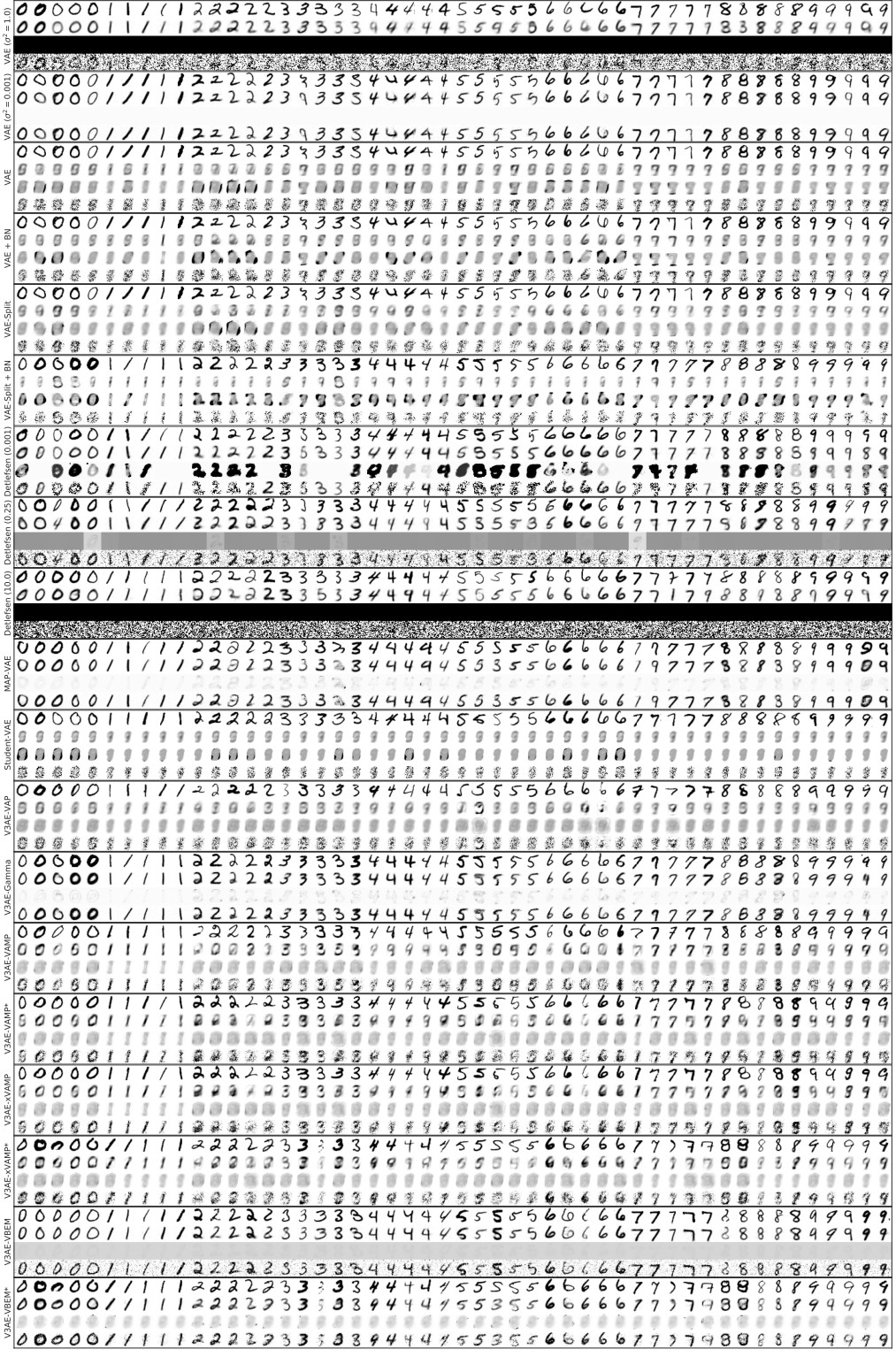

Figure 10: VAE PPC visualization for MNIST: The rows within a subplot from top to bottom are randomly selected test data followed by the local reconstruction distribution's mean and variance and a sample from it. Pixel values are clamped to $[0, 1]$, when PPC values exit this interval. Darker regions of variance images denote areas of higher predictive variance.

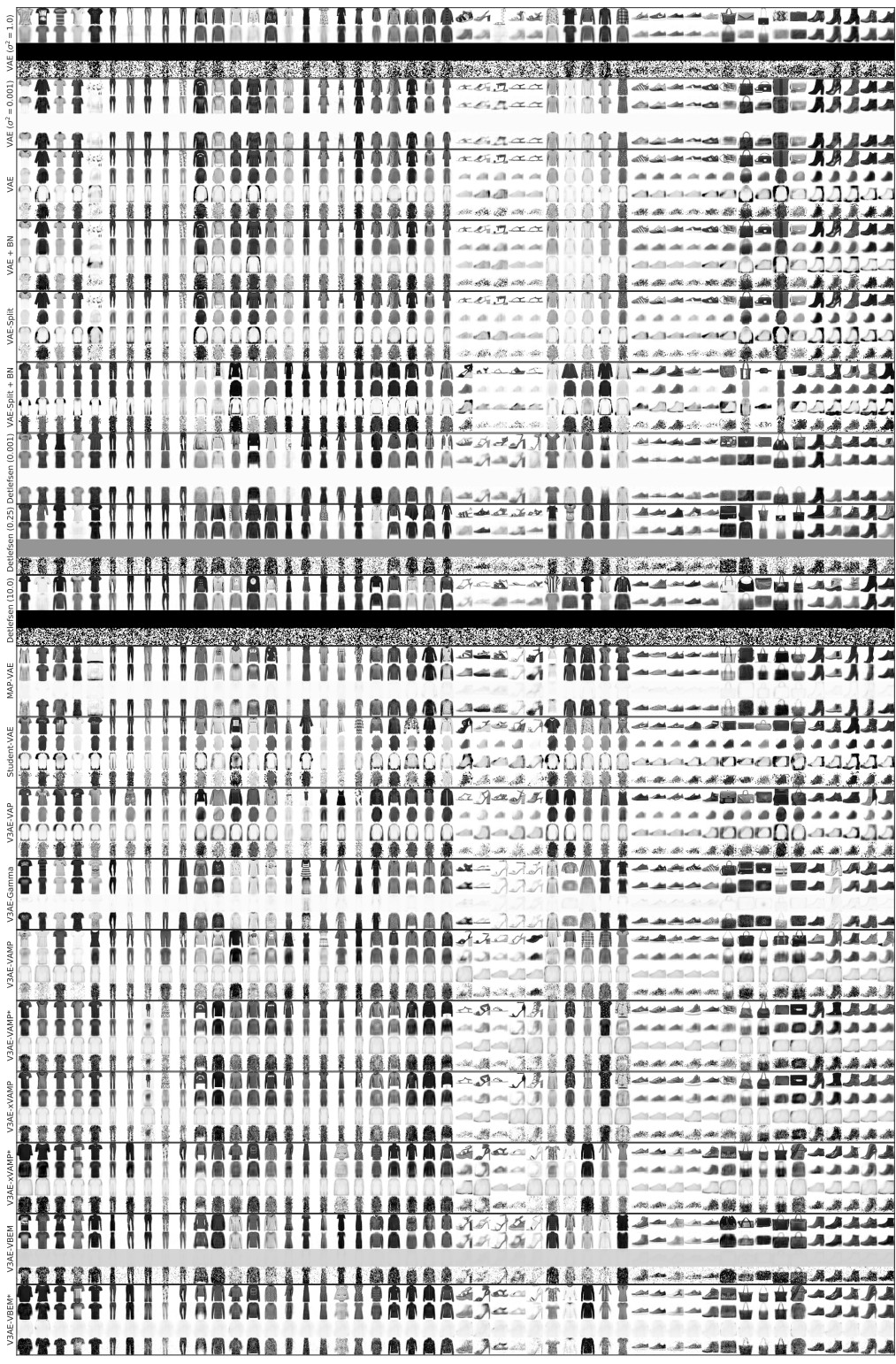

Figure 11: VAE PPC visualization for Fashion MNIST: The rows within a subplot from top to bottom are randomly selected test data followed by the local reconstruction distribution's mean and variance and a sample from it. Pixel values are clamped to $[0, 1]$, when PPC values exit this interval. Darker regions of variance images denote areas of higher predictive variance.

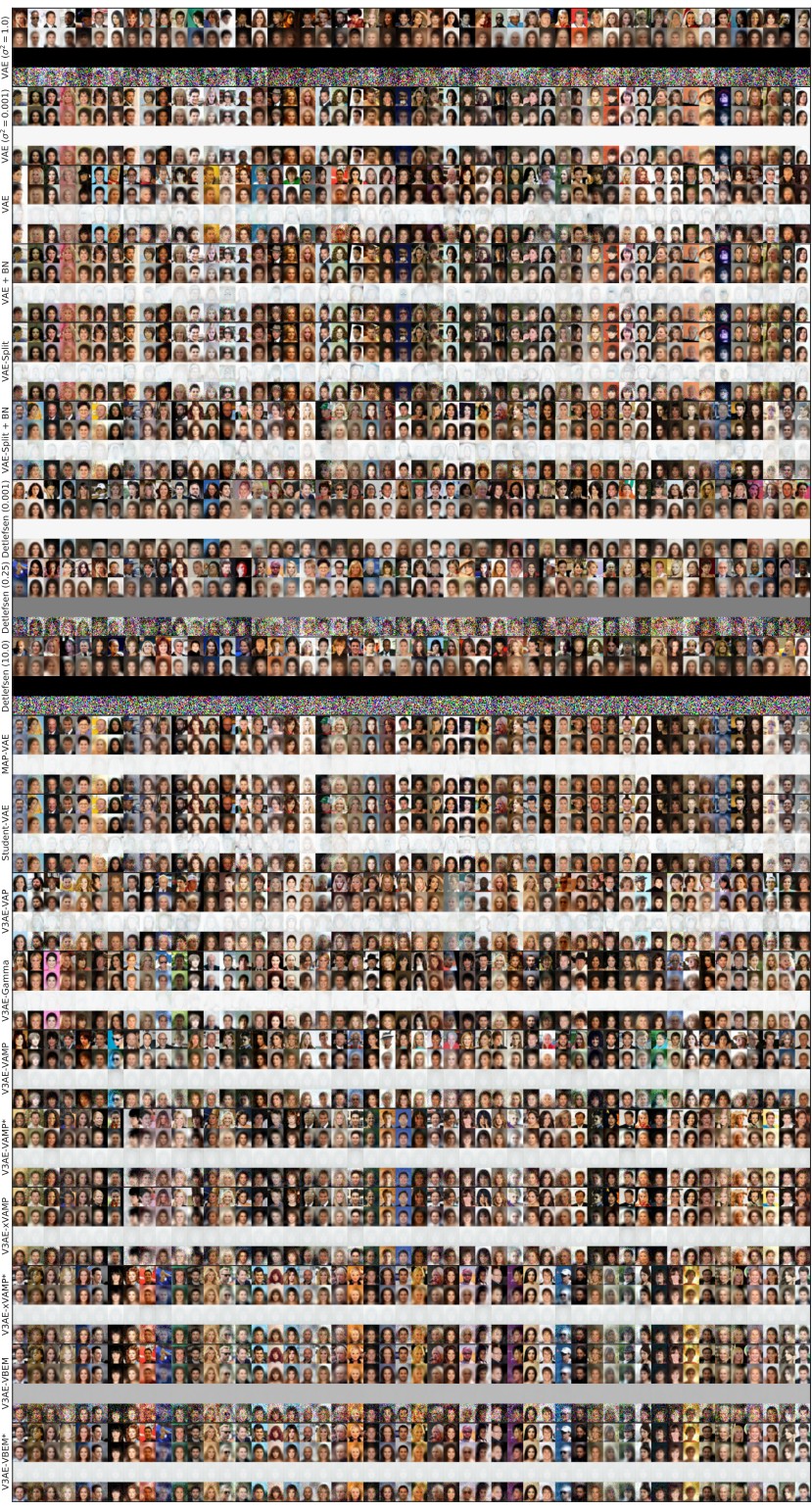

Figure 12: VAE PPC visualization for Celeb-a: The rows within a subplot from top to bottom are randomly selected test data followed by the local reconstruction distribution's mean and variance and a sample from it. Pixel values are clamped to $[0, 1]$, when PPC values exit this interval. Variance images are inverted such that lighter regions have lower predictive variance.

