# OpenReview forum: "Variational Variance: Simple and Reliable Noise Variance Parameterization"
_TMLR — Rejected by TMLR_

### Review · Reviewer_NaiR · 2022-07-19

**Summary Of Contributions:**

This paper addressed the problem of poor uncertainty quantification in variational Bayesian deep learning, specifically in the case where the conditional mean and variance of a Gaussian are functions parameterized by neural networks. In addition to bad uncertainty quantification of current variational methods based on parameter maximization, the authors argue that current methods often have poor optimization behavior due to numerical instability of parameters for observations with isolated covariates. The authors propose to address these issues using an empirical Bayes approach to regularize and posterior predictive checks to test for poor optimization performance. Numerical experiments are used to validate improved predictive performance and more reliable optimization.

**Broader Impact Concerns:**

I have no broader impact concerns

**Requested Changes:**

I view all the following changes as critical:

1. Restructure the paper to clearly delineate: the setting (both regression and unsupervised learning), previous work and its limitations, hypothesized problems, novel methods that solve these problems, and numerical experiments.
2. Add additional numerical experiments that validate the hypothesized problems are in fact the source of the observed issues. E.g., in the toy data experiments, what happens as the covariate of the outlier observation gets farther aways from the rest of the data? Do the gradients behave poorly in the way the authors hypothesize? Similar supporting experiments should be provided for the VAE setting.
3. Replace tables with graphical representations of the results


**Strengths And Weaknesses:**

Strengths:

1. Uncertainty quantification for DL methods is an important topic and replacing point estimation methods with a “more Bayesian” solution makes a lot of sense.
2. The inclusion of detailed experiments for both supervised and unsupervised problems is welcome.

Weaknesses:

1. It is hard for me to disentangle the precise contributions of the work and what made it different from previous work.
2. The experimental results do not fully support the two hypotheses given at the top of p. 2 (“mean network flexibility predicates MLE optimization instability” and “ we suspect a model that attempts placing a Dirac density on $Y = y_{ij} \mid X = x_i$ will experience exploding gradients that prohibit $\mu(x_i)$ from converging to $y_i$, thus forcing the model to use high variance”).
3. The innovation or even the precise use of posterior predictive checks is never clearly described.
4. The huge tables of numbers make it challenging to interpret the numerical results. As a rule, tables should only be used to summarize a small number of results.
5. The VAE section is particularly hard to read because it mixes together background/set-up, methods, discussion of implementation issues in code from another paper, and numerical results.

---

> ### Author Response · Authors · 2022-08-05
> **NaiR to Reviewer NaiR**
>
> *It is hard for me to disentangle the precise contributions of the work and what made it different from previous work.*
>
> Disentangling our contributions was a common complaint among the reviewers. We reworked the introduction to better differentiate us from existing work. Our primary and secondary contributions are now clearly listed.
>
> *The experimental results do not fully support the two hypotheses given at the top of p. 2 (“mean network flexibility predicates MLE optimization instability” and “ we suspect a model that attempts placing a Dirac density on Y=yij∣X=xi will experience exploding gradients that prohibit μ(xi) from converging to y, thus forcing the model to use high variance”).*
>
> This paragraph clearly needed some work based on the aggregated reviews. This discussion is meant to serve as background material and then explain how our solution has certain advantages over Takahashi et al. and Detlefsen et al. Takahashi et al. experimentally demonstrate how the zero-variance (a Dirac density) tendency leads to learning instability. Therefore, we did not feel the need to repeat these experiments. We merely want to point out that the zero-variance problem disappears whenever the mean network is not flexible enough to pass through an arbitrary $x_i, y_i$ directly (thus variance will never tend toward zero). Clearly, $x_i,y_i$ having a nearby neighbor $x_j,y_j$ will limit a mean network’s ability to pass through both simultaneously. Detlefsen et al. recognize this effect as part of their proposal to include $x_i$’s k nearest neighbors. We updated the introduction to make this more clear.
>
> Despite discussing the effect of nearby neighbors, Detlefsen et al.'s toy experiments did not simulate a lack of neighbors. We modified their toy experiment to simulate an $x_i, y_i$ without neighbors at x=7.5. The heteroscedastic Normal (Nix & Weigend, 1994) and Student (Takahashi et al. 2018) models both fail to converge on this point suggesting that while Takahashi et al. may have improved learning stability, their method does not produce a sensible predictive mean estimate–they never examine the quality of their mean predictions in their manuscript. At the request of another reviewer, we added a homoscedastic Normal (equivalent to minimizing MSE and then setting the global variance to the MSE) and found that it converges on the point x=7.5; this suggests that the mean network is flexible enough to converge on this isolated point. Thus, we can blame the Normal (Nix & Weigend, 1994) models convergence failure on the fact that we are jointly optimizing separate mean and variance networks simultaneously. We added language to better discuss how and why the various models succeed or fail in the toy experiments.
>
> *The innovation or even the precise use of posterior predictive checks is never clearly described.*
>
> Thank you for your suggestion–you were not alone in this request. We added a list of PPC definitions for regression and VAEs.
>
> *The huge tables of numbers make it challenging to interpret the numerical results. As a rule, tables should only be used to summarize a small number of results.*
>
> We are actively working on a graphical representation, but so far the plots have been too busy since we have ten datasets, four metrics, and eleven models with vastly varying numerical scales. Do you have any suggestions?
>
> *The VAE section is particularly hard to read because it mixes together background/set-up, methods, discussion of implementation issues in code from another paper, and numerical results.*
>
> Thank you for your feedback. We reorganized this section to better delineate the discussion.

---

> > ### Comment · Reviewer_NaiR · 2022-08-09
> > **Reply to author response and revised manuscript**
> >
> > Thanks to the authors for the substantial improvements to the clarity and readability of the manuscript.
> >
> > My major concern is now the use of so-called posterior predictive checks (PPCs) and their use in evaluating the competing models – which also relates issues around the use of tables to compare performance. Let $\boldsymbol{Y}$ denote all the observed responses. A PPC generically takes the form
> >
> > $\mathrm{PPC}(\mathcal{D}) = \mathbb{E}_{\tilde{\boldsymbol{Y}}}[\Delta(T(\boldsymbol{Y}), T(\tilde{\boldsymbol{Y}}))]$,
> >
> > where the expectation is with respect to the posterior predictive of the responses (conditioned on the covariates), $T$ is a statistic (e.g., mean, MSE, avg log-likelihood), and $\Delta$ is a distance function. For example, the PPC p-value uses a real-valued $T$ and $\Delta(t, \tilde{t}) = \mathbb{I}(t > \tilde{t})$. The “PPCs” described in Sections 3.2 and 4.1 are not of this form. However, there is recent work on held-out versions of PPCs [1,2]. I expect that using an actual PPC or one of the held-out versions described in those papers would lead to more interpretable and concise diagnostics. I would recommend the held-out versions since we should expect them to work much better for flexible models (see, e.g., the experiment in Section 5.4 of [1]). I would also suggest using a p-value as it is more interpretable. Some kind of categorical plot can then be used to concisely display all the p-values across models and datasets. Separate plots can be used for each statistic $T$.
> >
> > As it is, the manuscript does not use the “PPCs” to show that some models are well-calibrated (or at least better calibrated) in an interpretable way. Switching to a predictive p-value will help but I would also like to see the same PPC-type method used in the toy example to demonstrate how it can detect which models better fit the mean and are better calibrated.
> >
> > A few other more minor comments that need to be addressed:
> > * In the first sentence of Section 1, the Valdenegro-Toro reference should be parenthetical
> > * The parenthetical “e.g.” in the discussion of underrepresented groups at the top of p. 2 doesn’t make sense.
> > * After eq. (1): I think “regulate” should be “regularize”
> > * The discussion of the VB interpretation of eq. (1) is incoherent. The KL between the uniform prior on $\mu_i$ and $q$ is $\infty$ because the distributions are mutually singular (it’s not an issue of normalization).
> > * After eq. (2) there shouldn’t be a new paragraph start
> > * At the end of the same paragraph “variational posterior predictive” shouldn’t be italicized
> > * At the end of section 2, what does “variational variance” mean?
> > * I’m not familiar with the usage of “ablating” in Section 3
> > * For the experiments in 3.2, make sure to normalize the data based only on the training data – not using the training and testing data together
> > * Make sure references are up-to-date. For example, the Kendal & Gal and Kingma & Ba papers are published, as are a number of others.
> >
> > ### References
> >
> > [1] G. E. Moran, D. M. Blei & R. Ranganath. Population Predictive Checks (2022). arXiv:1908.00882v5 [stat.ME].
> >
> > [2] J. Li & J. H. Huggins. Calibrated Model Criticism Using Split Predictive Checks (2022). arXiv:2203.15897v2 [stat.ME].

---

> > > ### Author Response · Authors · 2022-08-14
> > > **Further Response**
> > >
> > > Thank you for your response. We appreciate your desire to have PPCs with a test statistic. Unfortunately, this would require a lot of work and additional computation (rerunning all of our experiments) as we did not save the intermediary information required to do so (e.g. model checkpoints). Your suggested definition of a PPC is generally used to determine if a single model is well calibrated. However our main interest is to compare fit across models.  Your proposal to compare p-values could be problematic if multiple models have p-values near zero or one—in practice, a value 0.001 is not much weaker than 0.00001 (Gelman et al., 2004). We compare posterior predictive performance between models such that our PPC takes a different form,
> > >
> > > $$\text{PPC}(\mathcal{D}) = \mathbb{E}[\Delta(T(Y,\tilde{Y}_\text{model 1}), T(Y,\tilde{Y}_\text{model 2}))]$$
> > >
> > > where the expectation is over both optimization randomization (e.g. initialization, batch selection) and the posterior predictive distributions of the two models. Gelman et al. (2004) introduce posterior predictive checking with:
> > >
> > > “If the model fits, then replicated data generated under the model should look similar to
> > > observed data. To put it another way, the observed data should look plausible under
> > > the posterior predictive distribution. This is really a self-consistency check: an observed
> > > discrepancy can be due to model misfit or chance.”
> > >
> > > Take our "PPC" that most closely matches Gelman's definition, the RMSE of a predictive sample, where we compare samples from the predictive distribution to the target response. While we don’t test for significance between the samples and the target response, we do test for significant differences in RMSE between two models. We feel our approach is well motivated since it is possible for two models to both have statistically acceptable (or unacceptable) PPC performance under your hypothesis test, while one of the models performs significantly better than the other. In either case, we clearly want to pick the better model. Thus, we feel that we are faithful to Gelman's definition of PPCs in our approach that compares between two models. Consider for example the VAE experiments. Sampling performance of the VAE with a fixed variance of 0.001 (the corresponding standard deviation is much less than the discretization of pixel values in [0,255]), should represent the best possible sampling performance. Here, we don’t care about its individual PPC’s p-value, but rather how other models deviate from this particular model. In our opinion, the “best” heteroscedastic model is one whose predictive mean performance is just as well calibrated as a model that only fits the mean (i.e. its predictive mean calibration is statistically indistinguishable from this fixed variance baseline). While we don’t achieve this goal, we make progress towards it.
> > >
> > > With regards to your minor points, we agree with most of them and will correct them. We apologize that you found our discussion of equation 1 incoherent, but we cannot discern from your comment what you would like improved. We specifically discuss the infinite KL divergence that you mention and note that it is constant with respect to the variational parameters and can thus be ignored. In section 3, "ablating" refers to our removal of something. Our main proposal is to use Variational Bayes and the Normal and Student models are important baselines we consider. The Normal model is MLE and removes the Bayesian treatment. The Student model directly optimizes the log predictive likelihood of our proposal and thus removes the variational aspect while remaining Bayesian with respect to precision. We normalize both the covariate and response variables using only training data. Please see experiments_regression.py to confirm.

---

> > > > ### Comment · Reviewer_NaiR · 2022-08-18
> > > > **PPCs**
> > > >
> > > > Thanks for your response.
> > > >
> > > > ### Evaluation
> > > >
> > > > Regarding your current evaluation metrics, which are not PPCs and need to be called something else. Using average log predictive likelihood and RMSEs are perfectly reasonable metrics. However, I find the bias of the predictive variance to be more suspect. Just because on average the predictive variance is correct doesn't mean any of the predictive variances are any good. Overall, as currently presented in the table, I don't find that the current metrics provide compelling evidence that the proposed methods are better calibrated. It might be true but you need to do some more work to make a strong case. And in any case, I don't see the use of these metrics as a meaningful contribution of the manuscript.
> > > >
> > > > Regarding the comparison of models, you could use something like the [posterior predictive null check](https://projecteuclid.org/journals/bayesian-analysis/advance-publication/The-Posterior-Predictive-Null/10.1214/22-BA1313.full). Alternatively, you could use two-sample tests or some other appropriate hypothesis testing framework to rigorously provide a partial ordering on method performance.
> > > >
> > > > ### KL divergence
> > > >
> > > > The problem with your discussion of eq. (1) is not the fact that the uniform prior can't be normalized. It's that you are calculating the KL divergence between mutually singular distributions (one concentrated on $\\{0\\}$, the other on $\mathbb{R} \setminus \\{0\\}$), which is incoherent.
> > > >
> > > > ### Ablating
> > > >
> > > > Please restate without using the term "ablating" to make it more clear what you are saying.

---

> > > > > ### Author Response · Authors · 2022-09-07
> > > > > **Following up**
> > > > >
> > > > > Thank you for your insightful comments. We chose to show variance bias since we observed the baseline models using high variance to explain poor mean convergence and we wanted to highlight this failure mode. We completely agree that near-zero bias does not mean the predictive variance is any good. We fortunately logged the RMSE of the predicted variance,
> > > > >
> > > > > $$\Big(|D_\text{test}|^{-1}\sum\limits_{(x_*,y_*)\in D_\text{test}} \big(\text{var}[Y_*|X_* = x_*] - \big(E[Y_*|X_* = x_*] - y_*\big)^2\big)^2\Big)^\frac{1}{2},$$
> > > > >
> > > > > when we ran the original experiments. We were able to include this equation and include the corresponding table in the appendix (without needing to rerun experiments). We will upload an updated manuscript shortly.
> > > > >
> > > > > With regards to PPC nomenclature, we are happy to call them something else. How does "a pair-wise comparison of PPC test statistics", sound? We don't want to adopt posterior predictive null checks because our pre-print precedes the linked paper and this would require significant work on our end. Our understanding is TMLR emphasizes correctness, and we feel our analysis is correct.
> > > > >
> > > > > With regards to you KL divergence comment, we can make the approximation exact by instead treating both the variational dirac density and uniform prior as their corresponding discrete distributions over the 32-bit floating point system that we use in implementation.

---

### Review · Reviewer_TycY · 2022-07-20

**Summary Of Contributions:**

In this paper, the authors propose a variational treatment for the noise parameter in the heteroscedastic Gaussian likelihood models. The authors analyze their proposed variational family under different priors and also propose a few new priors. The authors offer extensive experiments and prior predictive checks.


**Broader Impact Concerns:**

In the introduction, authors talked about how the optimization issues may be more relevant to underrepresented groups. However, in the paper, I did not really find an analysis on what was the impact of their proposed method in such settings. Maybe the authors would want to address that in a broader impact section (and possibly in their paper.)

**Requested Changes:**

Please, refer to the strength and weakness section for detailed comments.

**Strengths And Weaknesses:**

# Strength

### Extensive experiments

Authors provide extensive experimental evidence to corroborate their claims.

### Several contributions

It seems to me that there are several small contributions in this paper. However, how the combination of these contributions influences the final output is a bit unclear to me (see section on Weakness for detailed comments.)

Overall, I think the paper has merit and can be a welcome addition to the literature; however, I think the current presentation can improve. I mostly have issues with writing that made it hard to put the contributions in proper context for me. I details my comments on weaker side of the paper and encourage the authors to offer explanations.

# Weakness

## Claims

### ELBO stabilization

On page 2, before equation 2, the authors talk about how there is a one to one correspondence between log-likelihoods and the KL term which leads to ELBO stabilization. However, I fail to see how this is unique to the proposed method. Any other heteroscedastic method will have the same effect?

### Effect of nearby neighbors

On page 2, authors talk about the effect a nearby neighbor can have on the optimization. However, I did not see any experiment in the paper that establish this hypothesis. I think the toy experiments were supposed to bring that out. However, I am not sure if the authors were able to establish that as clearly. Also, Detlefsen et. al. 2019, in section 3.1 of their paper, talk about the effect of not having nearby neighbors in a minibatch. I fail to see how the hypothesis in this paper is different from their observations.

## PPCs

The authors also claim that they propose several prior predictive checks. Are there any novel prior predictive checks that the authors use in their analysis? If so, I could not really find it within the manuscript. If not, it seems a bit misleading to me that authors write the use of standard prior predictive checks as a contribution.

## Unclear writing

I found the paper hard to read and believe it can improve from a rewrite.

### General Writing

The narration of the paper seems a bit confusing to me. What are the core contribution of the paper? Is it the proposed variational family? Is it the proposed priors?

Writing in section 2 is also a bit confusing. A lot of approaches are introduced without establishing clear the context of the problem. There is MLE, there is black-box VI, there is variational EM, there is and there is empirical Bayes. Then. when we go to the experiments on variational autoencoders in section 4, there is more notation and formulations.

### Analysis

It is a bit hard to find the analysis for the variational family uncoupled from the choice of the prior. How is the proposed formulation different and better then the existing variational families/parameterization? Maybe the experiments are there (based on the extensive set of numbers in the paper); however, they are very hard to make sense of. Can the authors comment using some other variational parameterization with the proposed priors? In essence, if we used some other parameterization with priors from Table 1, are the results worse?

**Figure 1:** why do training dots look different for different columns and rows? I assuming they are randomly selected out of the 20 trials that authors ran. Why not standardize the datasets for visualization?

### Contributions

On page 2, first paragraph, the authors claim they extend the study of Takahashi et. al. 2018 in several ways. The rest of the section reads as the conjectures and suspicions authors have. I was confused by the purpose of this paragraph.

In general, the exact contributions of the paper are a bit unclear to me. Is it the proposed variational treatment of the precision term? Is it the priors? Is it the PPCs? Is it all of them? If so, I think we need some ablation studies to understand the impact of the first two. For the latter, I need a clear description of the proposed PPCs from the authors.

### Details

Overall, I think the paper can benefit from a fresh rewrite. Currently, I think several details are mixed with the parts that cover the core contributions of the paper. Consider the first paragraph in section 3; the authors mention details about the parameterization of the covariance. I think these details can pushed to appendix and that improve the readability of the manuscript. Another instance, is the last paragraph on page 6; there are a lot of details about how authors corrected some code; other instances are the last paragraph on page 3 and second paragraph on page 4 (baselines of another paper).

---

> ### Author Response · Authors · 2022-08-05
> **Response to Reviewer TycY (part 1)**
>
> *On page 2, before equation 2, the authors talk about how there is a one to one correspondence between log-likelihoods and the KL term which leads to ELBO stabilization. However, I fail to see how this is unique to the proposed method. Any other heteroscedastic method will have the same effect?*
>
> You are absolutely correct that this is true for any variational treatment of heteroscedastic variance parameters. We simply are the first, to the best of our knowledge, to use such a variational treatment to improve reliability of a network’s heteroscedastic variance estimate and view the 1:1 relationship as a “feature” of our approach since it cannot be overwhelmed by additional data (i.e. our regularization scales to datasets of any size). We clarified this in the relevant text.
>
> *On page 2, authors talk about the effect a nearby neighbor can have on the optimization. However, I did not see any experiment in the paper that establish this hypothesis. I think the toy experiments were supposed to bring that out. However, I am not sure if the authors were able to establish that as clearly. Also, Detlefsen et. al. 2019, in section 3.1 of their paper, talk about the effect of not having nearby neighbors in a minibatch. I fail to see how the hypothesis in this paper is different from their observations.*
>
> We did not intend for our hypotheses to differ. The section you are referring to in Detlefsen et al. discusses the problem of lacking nearby neighbors in order to motivate their “locality sampler” (proposed in their section 3.2), where they include a batch member’s k nearest neighbors. However, their toy experiment does not simulate the effect of lacking a nearby neighbor in the dataset; this motivated our modification that does simulate a data point without nearby neighbors. In our version of the experiment, we show that parameterizing a heteroscedastic Normal model (Nix & Weigend, 1994) and a heteroscedastic Student model (Takahashi et al., 2018) with neural networks will prohibit the mean from converging on an isolated point. At the request of another reviewer, we added a homoscedastic Normal (equivalent to minimizing MSE and then setting the global variance to the MSE) and found that it converges on the point x=7.5; this suggests that the mean network is flexible enough to converge on this isolated point, but fails to do so when jointly optimizing a separate variance network. Takahashi et al. discuss the zero-variance problem to motivate their Student model. They conduct a variety of experiments that show their proposed Student model improves loss stability and attains higher log likelihoods than the Normal model. Yet, they never examine the quality of their method’s predictive mean and variance. Looking at the Student model for the UCI and VAE experiments we see that even though the Student model often has the best log likelihood its mean and variance are not well calibrated. Critically and in contrast to Detlefsen et al., our approach does not rely on finding x_i’s nearest neighbors (which requires defining a distance metric for x). As we suspected, their nearest neighbor solution breaks down in higher dimensions for VAEs where we find that their predictive variance estimates are largely insensible. We reworked our introduction and toy experiment sections to better address these points.
>
> *The authors also claim that they propose several prior predictive checks. Are there any novel prior predictive checks that the authors use in their analysis? If so, I could not really find it within the manuscript. If not, it seems a bit misleading to me that authors write the use of standard prior predictive checks as a contribution.*
>
> We agree that our sentence, “First, we propose several posterior predictive checks (PPCs) (Gelman et al., 2013) to critique predictive mean and variance, allowing us to detect MLE optimization failures, which we indeed find alongside high model likelihoods” is misleading and does not convey what we meant to communicate. Rather, our novelty is the adoption of standard predictive checks as a way to diagnose poorly calibrated heteroscedastic models that rely on amortized neural networks for their parameter maps. To the best of our knowledge, we are the first to do so for this class of models. Takahashi et al. consider log likelihoods alone. Detlefsen et al. examine a model’s log likelihood and RMSE and, when the true variance is known, the model’s variance. We consider log likelihoods, RMSE, variance (even when the true variance is unknown), and samples from the predictive distribution. We cleared up the above language in the introduction and added a list of PPC definitions to the regression and VAE sections, respectively.
>
> *I found the paper hard to read and believe it can improve from a rewrite.*
>
> Please find that we rewrote significant portions of the manuscript.

---

> ### Author Response · Authors · 2022-08-05
> **Response to Reviewer TycY (part 2)**
>
> *The narration of the paper seems a bit confusing to me. What are the core contribution of the paper? Is it the proposed variational family? Is it the proposed priors?*
>
> Thank you for this criticism; this was a common complaint among the reviewers. We reworked the introduction to better differentiate us from existing work. Our primary and secondary contributions are now clearly listed.
>
> *Writing in section 2 is also a bit confusing. A lot of approaches are introduced without establishing clear the context of the problem. There is MLE, there is black-box VI, there is variational EM, there is and there is empirical Bayes. Then. when we go to the experiments on variational autoencoders in section 4, there is more notation and formulations.*
>
> Thank you for your suggestion. We reworked significant portions of our manuscript to draw better connections between the methods.
>
> *It is a bit hard to find the analysis for the variational family uncoupled from the choice of the prior. How is the proposed formulation different and better then the existing variational families/parameterization? Maybe the experiments are there (based on the extensive set of numbers in the paper); however, they are very hard to make sense of. Can the authors comment using some other variational parameterization with the proposed priors? In essence, if we used some other parameterization with priors from Table 1, are the results worse?*
>
> Thank you for your suggestion, we chose to keep the variational family as a Gamma distribution with amortized parameter maps because it is conjugate prior for a Normal with unknown precision. This fact admits analytic integration such that the predictive distribution is a Student t. At one point, we used a LogNormal instead of the Gamma. However, this consistently performed worse and was computationally much more expensive since it required Monte-Carlo integration to approximate the posterior predictive. And as you point out, we already have too many numbers! Furthermore, by keeping the variational family as a Gamma, there is a subtle relationship between the Normal, Student t, and our variational methods. Optimizing the Student model is the same as optimizing our variational posterior predictive directly. In this light, both the Student and our variational method are Bayesian treatments of a Normal with unknown precision. We reworked section 3 to better discuss the connections between models.
>
> *Figure 1: why do training dots look different for different columns and rows? I assuming they are randomly selected out of the 20 trials that authors ran. Why not standardize the datasets for visualization?*
>
> Thank you for pointing this out. We actually select the best run for each model. We did so to emphasize that the heteroscedastic Normal model (Nix & Weigend, 1994) and the heteroscedastic Student t model (Takahashi et al., 2018) always fail to converge on the isolated point across all our trials. We added language to this effect.
>
> *On page 2, first paragraph, the authors claim they extend the study of Takahashi et. al. 2018 in several ways. The rest of the section reads as the conjectures and suspicions authors have. I was confused by the purpose of this paragraph.
> Thank you for your critique.*
>
> We agree this portion of the introduction needed a significant rewrite. We hope you find the reworked introduction as an improvement.
>
> *In general, the exact contributions of the paper are a bit unclear to me. Is it the proposed variational treatment of the precision term? Is it the priors? Is it the PPCs? Is it all of them? If so, I think we need some ablation studies to understand the impact of the first two. For the latter, I need a clear description of the proposed PPCs from the authors.*
>
> Disentangling our contributions was a common complaint among the reviewers. We reworked the introduction to better differentiate us from existing work and clearly identify our contributions.
>
> We need to better emphasize our ablation studies. Our proposal is a Bayesian treatment of unknown precision that uses amortized variational inference. Ablating the Bayesian treatment, gives us the Normal model. Ablating amortized variational inference gives us the Student model, which optimizes our methods variational posterior predictive directly. For the priors, we ablate their effect by setting the KL term to zero (via the VAP prior). We reworked section 3 to better discuss these connections.

---

### Review · Reviewer_Xf8f · 2022-07-20

**Summary Of Contributions:**

This paper introduces a scheme for fitting neural networks for heteroscedastic regression tasks and VAEs which is necessary for adequately modeling prediction uncertainty. Previous work in the area has shown that fitting these sorts of networks can be unstable. The proposed method treats the variances variationally and uses an Empirical Bayes style method to find the appropriate priors. Comparisons of fit are done using posterior predictive checks. Experiments focus on regression tasks and VAEs on standard benchmarks and show promising results.

**Broader Impact Concerns:**

No broader impact concerns.

**Requested Changes:**

Please address the noted weaknesses. In particular the interpretation of $\mu(\cdot)$ and the PPCs.

**Strengths And Weaknesses:**

Strengths:
- Strong empirical results in appropriate experiments
- Straightforward, simple approach to the problem: treat the variance/precision variationally

Weaknesses:
- The only presented example with ground truth is for heteroskedastic noise. How does this method perform in the situation where the noise is homoskedastic? How much loss in efficiency is there compared to using a homoskedastic approach?
- More detail on the role and interpretation of the $\mu(\cdot)$ network. We are only being “Bayesian” about the $\lambda_i$, so what is the interpretation of $\mu(\cdot)$? From a Bayesian perspective is this network to be interpreted as a prior (point-mass priors on the means) and also tuned in an Empirical Bayes manner? Are we then sampling out of the prior at prediction time?
- What are the computational/time costs of having three neural networks ($\mu, \alpha, \beta$) rather than only two as in the MLE set-ups?
- No reference or derivation of eq. (2)
- PPCs are put forward as a major contribution of this article, but there is limited discussion of the PPCs themselves (there is discussion of the PPCs when comparing the methods). Most discussion of PPCs (for the regression experiments) is in 3.2, after the toy experiments are discussed. Perhaps a subsection on evaluation/PPCs would structurally make more sense.
- Table 3: just above this table it says “Comparing ELBOs (including tighter impor- tance weighted ELBOs (Burda et al., 2015)) across these methods is problematic” then Table 3 proceeds to report the ELBOs.

---

> ### Author Response · Authors · 2022-08-05
> **Response to Reviewer Xf8f**
>
> Thank you again for your feedback. Please find our responses inline.
>
> *The only presented example with ground truth is for heteroskedastic noise. How does this method perform in the situation where the noise is homoskedastic? How much loss in efficiency is there compared to using a homoskedastic approach?*
>
> Thank you for your suggestion. We added an additional toy experiment where the underlying noise variance is homoscedastic and still assumes an isolated point at x=7.5 (as in Figure 1) to our appendix (newly created Figure 5). We also implemented a homoscedastic Normal model, where we set the variance to the mean of squared errors of the training points. For plot aesthetics, we replaced the Detlefsen plot (i.e. the one with their bug) with the homoscedastic model for Figure 1 and 4. We created a new appendix figure comparing before and after our bug fix. In summary, the homoscedastic model converges on the isolated point, confirming that jointly optimizing a variance network is the cause for the heteroscedastic Normal model’s inability to converge on the same isolated point. As expected, the homoscedastic model, as we implemented it, estimates the variance to be the E[true var(y|x)] evaluated with respect to the training distribution.
>
> *More detail on the role and interpretation of the μ(⋅) network. We are only being “Bayesian” about the λi, so what is the interpretation of μ(⋅)? From a Bayesian perspective is this network to be interpreted as a prior (point-mass priors on the means) and also tuned in an Empirical Bayes manner? Are we then sampling out of the prior at prediction time?*
>
> Thank you for your astute inquiry. Your interpretation is one of several valid interpretations. We added your interpretation along with two others in the second paragraph of section 2. First, we are simply performing MLE for the mean and Variational Bayes for the precision simultaneously. There is no rule that forbids combining inference methods. For example, using MCMC and Variational Bayes simultaneously has benefitted topic models (Minmo et al. 2012). Second is the fully Variational Bayesian interpretation is that we place a uniform prior over the reals on $\mu_i$ and define $q(\mu_i) \triangleq \delta(\mu(x_i))$. The mean’s resulting KL divergence subtracts a constant, negative infinity from the ELBO, which can be ignored since it has zero gradient w.r.t. to the mean network's parameters.
>
> *What are the computational/time costs of having three neural networks (μ,α,β) rather than only two as in the MLE set-ups?*
>
> If all networks are identical except the final activation function (and we ignore the final activation functions’ costs), then the computational cost of three networks is simply 1.5 times the cost of two networks. Of course, one could merge multiple networks into one and have a multi-headed output. If a multi-headed output occurs at the final layer, then each scalar output adds only an inner product (hidden output dotted with parameters) and the negligible activation function. We did not consider this more compact architecture in order to maintain parity with Detlefsen et. al and Takahashi et. al. We added a blurb to this effect in section 3.
>
> *No reference or derivation of eq. (2)*
>
> Thank you for pointing this out. We think the actual problem is that we forgot to mention that the variational family, $q(\lambda|\alpha(x),\beta(x))$ is a Gamma distribution. Regardless, we should not assume that everyone knows that taking the expectation of a Normal pdf with respect to a Gamma-distributed precision (i.e. integrating over unknown precisions) results in a Student’s T distribution. We will clarified the text and added a citation to Bishop’s machine learning textbook that has the integral derivation in section 2.3.7.
>
> *PPCs are put forward as a major contribution of this article, but there is limited discussion of the PPCs themselves (there is discussion of the PPCs when comparing the methods). Most discussion of PPCs (for the regression experiments) is in 3.2, after the toy experiments are discussed. Perhaps a subsection on evaluation/PPCs would structurally make more sense.*
>
> Thank you for your suggestion–you were not alone in this request. We added a list of PPC definitions for regression and VAEs.
>
> *Table 3: just above this table it says “Comparing ELBOs (including tighter impor- tance weighted ELBOs (Burda et al., 2015)) across these methods is problematic” then Table 3 proceeds to report the ELBOs.*
>
> Thank you for pointing this out. We included them for reference only. We happily removed them–we only added them based on a previous reviewer’s request.

---

### Author Response · Authors · 2022-08-02
**Thank you and please expect our response shortly!**

We thank all reviewers for their time and helpful feedback. We are still preparing our responses and a revised manuscript, which we anticipate providing in the next 36 hours.

---

### Decision · Action_Editors · 2022-09-16

**Recommendation:** Reject

**Comment:**

The authors propose training a heteroskedastic regression model in the overparameterized limit. In this setup, the learnable data variances can lead to training instabilities and poor uncertainty calibration since the model can fit every datum perfectly. They propose to treat the output variances in a Bayesian fashion as a solution, using amortized variational inference. The authors propose a variant of posterior predictive checks to study the calibration properties of the augmented model.

The paper discusses a relevant topic and has experiments from both supervised and unsupervised domains.

Unfortunately, the paper has two major weaknesses: (1) an unconventional definition of "posterior predictive check" whose usefulness for measuring calibration was doubted by the reviewers, and (2) an unclear writing style that makes the paper hard to read. Unfortunately, the majority of reviewers agreed that the paper requires substantial further editing in regards to clarity, as well as potentially additional experiments showing a more conventional version of posterior predictive check.